# Leveraging Separated World Model for Exploration in Visually Distracted Environments

**Kaichen Huang**[1,2*], **Shenghua Wan**[1,2*], **Minghao Shao**[1,2],
**Hai-Hang Sun**[1,2], **Le Gan**[1,2†], **Shuai Feng**[3], **De-Chuan Zhan**[1,2]

[1]School of Artificial Intelligence, Nanjing University, China
[2]National Key Laboratory for Novel Software Technology, Nanjing University, China
[3]School of Cyberspace Science and Technology, Beijing Institute of Technology, China
{huangkc,wansh,shaomh,sunhh}@lamda.nju.edu.cn,
{ganl,zhandc}@nju.edu.cn, fengshuai@bit.edu.cn

## Abstract

Model-based unsupervised reinforcement learning (URL) has gained prominence for reducing environment interactions and learning general skills using intrinsic rewards. However, distractors in observations can severely affect intrinsic reward estimation, leading to a biased exploration process, especially in environments with visual inputs like images or videos. To address this challenge, we propose a bi-level optimization framework named **Se**paration-assisted e**X**plorer (SeeX). In the inner optimization, SeeX trains a separated world model to extract exogenous and endogenous information, minimizing uncertainty to ensure task relevance. In the outer optimization, it learns a policy on imaginary trajectories generated within the endogenous state space to maximize task-relevant uncertainty. Evaluations on multiple locomotion and manipulation tasks demonstrate SeeX's effectiveness.

## 1 Introduction

Unsupervised learning has a rich history in computer vision and natural language processing, as demonstrated by methods such as [12, 23, 25, 9, 48, 14]. It leverages unlabeled, task-agnostic data to train models that can be quickly adapted to downstream tasks, addressing sample inefficiency. This approach has also gained traction in unsupervised reinforcement learning (URL) [35, 29, 36], where skills are developed through intrinsic motivation rather than external rewards. A key advantage of URL is that learned dynamics models can gather prior knowledge about the environment during exploration, minimizing the need for extensive interactions during adaptation to downstream tasks.

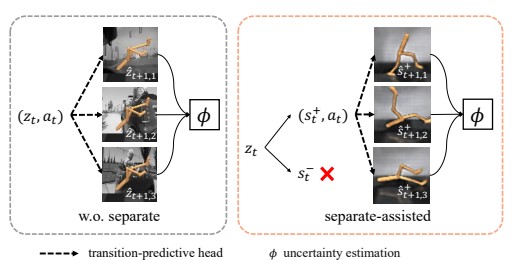

Figure 1: Comparison of traditional URL methods using a single world model (left) versus our separated world model (right).

A major challenge in unsupervised reinforcement learning arises from distractors, which significantly hinder learning in complex real-world environments. For example, a book-finding robot in a library must identify relevant objects (like books) while ignoring irrelevant ones (such as posters or people) to effectively complete its task. These distractors exponentially expand the original state space, creating numerous redundant states that impede efficient exploration [64]. As shown in Figure 1-Left, a single world model encodes both task-relevant and task-irrelevant information in the latent space. Even

---

[*]Equal contribution.
[†]Corresponding author.

38th Conference on Neural Information Processing Systems (NeurIPS 2024).

if the agent's state remains unchanged in $\hat{z}_{i+1,*}$, varying background distractors inflate uncertainty estimates. In contrast, as illustrated in Figure 1-Right, our proposed separation world model distinctly separates task-relevant information $s^+$ from task-irrelevant information $s^-$, allowing for accurate uncertainty estimation unaffected by distractors.

Despite its prevalence in real-world environments, this challenge has received limited attention in unsupervised reinforcement learning. Current unsupervised RL methods: spanning data-driven, knowledge-based, and competence-based approaches (see Section 6), remain vulnerable to redundant states, which can lead agents down unproductive exploration paths [69].

To address visual inputs with complex distractors, we propose a bi-level optimization framework called Separation-Assisted Explorer (SeeX). We extend URLB-pixels to environments with distractors, demonstrating that performance declines vary with task difficulty. Our method, SeeX, separates latent information in the world model into task-relevant and task-irrelevant components. We assume the task-irrelevant part captures transitions, using it to predict rewards and actions.

Our contributions are summarized as follows: **(I)** We introduce Separation-Assisted Explorer (SeeX), a new approach for tackling unsupervised RL tasks with complex distractors in observations. **(II)** We provide a theoretical analysis that formalizes URL with distractors, where the policy maximizes task-relevant uncertainty while the world model minimizes environmental uncertainty. **(III)** We demonstrate the outstanding performance of policies learned by SeeX across various locomotion and manipulation tasks with diverse visual distractors.

## 2 Preliminary

World models are highly effective for reinforcement learning (RL) tasks with visual inputs, performing well in both simulated [20, 21, 56] and robotics [67, 66] environments. In unsupervised settings, agents explore using intrinsic rewards during pre-training. This task-agnostic data facilitates fine-tuning through a world model. URLB-pixels [50] demonstrates that unsupervised methods combined with world models outperform model-free agents. Inspired by these results, we adopted a model-based framework in our approach.

Several studies, including URLB [33] and URLB-pixels [50], have created a unified framework for various unsupervised reinforcement learning (URL) methods, advancing the field. The benchmark consists of two phases: pre-training (PT), where agents explore without task-specific rewards, and fine-tuning (FT), where they apply knowledge gained during PT with limited interactions. While URLB-pixels effectively evaluates pixel-based performance, it does not address complex distractors in visual inputs. To fill this gap, we tested baseline methods on noisy video backgrounds and developed a novel approach to tackle this challenge.

Our focus lies in scenarios where agents must acquire skills from noisy visual observations, often disrupted by task-irrelevant videos. To model the dynamics under these conditions, we adopt the EX-BMDP framework, an adaptation of Block MDP. A Block MDP can be represented as a tuple $\mathcal{M} = (\mathcal{O}, \mathcal{Z}, \mathcal{A}, \mathcal{T}, \mathcal{R}, \mathcal{U})$, where $\mathcal{O}$, $\mathcal{Z}$ and $\mathcal{A}$ denotes the set of observations, latent states and actions respectively; the transition function $\mathcal{T} : \mathcal{Z} \times \mathcal{A} \to \Delta(\mathcal{Z})$ maps latent states and actions to probability distributions over next latent states; the reward function $\mathcal{R} : \mathcal{O} \times \mathcal{A} \to [0, 1]$ assigns reward to observation-action pairs; the emission function $\mathcal{U} : \mathcal{Z} \to \Delta(\mathcal{O})$ maps latent states to probability distributions over observations. The EX-BMDP is formulated as follows:

**Definition 2.1.** (Exogenous Block Markov Decision Process). An EX-BMDP is a BMDP such that the latent state can be decoupled into two parts $z = (s^+, s^-)$ where $s^+ \in \mathcal{S}^+$ is endogenous state and $s^- \in \mathcal{S}^-$ is the exogenous state. For $z \in \mathcal{Z}$ the initial distribution and transition functions are decoupled, that is: $\mu(z) = \mu_+(s^+)\mu_-(s^-)$, and $\mathcal{T}(z'|z, a) = \mathcal{T}_+(s^{+'}|s^+, a)\mathcal{T}_-(s^{-'}|s^-)$.

## 3 Bi-level Optimization for Exploration in Distracted Environments

The objective of the pre-training phase in URL is to learn a world model capable of handling downstream tasks with diverse reward functions. Following previous work [13, 8, 52], we formalize the pre-training phase as a minimax regret problem.

**Problem 3.1.** *(Regret Optimization for World Model) In the context of reward-free BMDP, consider a world model $W$ that defines the latent dynamics $\widehat{\mathcal{M}}^R$, which simulates the real dynamics $\mathcal{M}$*

*under a reward function R. Let $V_{\mathcal{M}^R}(\pi)$ denotes the expected value of policy $\pi$ under dynamics $\mathcal{M}$ with reward function R. The regret of policy $\pi$ is defined as $REGRET(\pi, \mathcal{M}^R) = V_{\mathcal{M}^R}(\pi^*) - V_{\mathcal{M}^R}(\pi), \pi^* = \arg\max_{\pi'} V_{\mathcal{M}^R}(\pi')$, which measures the performance gap between $\pi$ and the optimal policy. The optimization goal is to find the world model that is robust to different possible reward functions, minimizing the regret of world model policy under real dynamics:*

$$\min_W \max_R REGRET(\widehat{\pi}_R^*, \mathcal{M}^R), \widehat{\pi}_R^* = \arg\max_\pi V_{\widehat{\mathcal{M}}^R}(\pi) \tag{1}$$

In the absence of the reward function during pre-training, computing regret under specific reward functions becomes infeasible. To address this challenge, we propose transforming regret into a novel, reward-free objective using Simulation Lemma [28, 52]. Our proposition is outlined below.

**Proposition 3.2.** *Denote the learned latent dynamics in the world model as $\widehat{\mathcal{T}}$ and the true latent dynamics as $\mathcal{T}$. For any reward function R, the regret of the optimal world model is bounded by:*

$$\begin{aligned} REGRET(\widehat{\pi}_R^*, \mathcal{M}^R) \leq \frac{2\gamma}{(1-\gamma)^2} \Big[ &\mathbb{E}_{z,a \sim d(\pi_R^*, \widehat{\mathcal{M}})} \Big[ TV\big(\widehat{\mathcal{T}}(\cdot|z,a), \mathcal{T}(\cdot|z,a)\big) \Big] \\ &+ \mathbb{E}_{z,a \sim d(\widehat{\pi}_R^*, \widehat{\mathcal{M}})} \Big[ TV\big(\widehat{\mathcal{T}}(\cdot|z,a), \mathcal{T}(\cdot|z,a)\big) \Big] \Big] \end{aligned} \tag{2}$$

As outlined in Proposition 3.2, the optimal world model policy exhibits low regret if the latent state-action distribution of both $\pi_R^*$ and $\hat{\pi}_R^*$ in $\widehat{\mathcal{M}}$ is accurately captured by $\widehat{\mathcal{T}}$. However, during unsupervised pre-training, the reward function remains inaccessible, preventing us from directly obtaining these distributions. This is because the state-action distribution induced by $\pi_R^*$, and $\hat{\pi}_R^*$ is inherently dependent on the reward function. To address this challenge, we introduce an exploration policy $\pi^{expl}$, specifically designed to maximize the expected error (in terms of total variation distance) of the latent dynamics model:

$$\pi^{expl} = \arg\max_\pi \mathbb{E}_{z,a \sim d(\pi, \widehat{\mathcal{M}})} \Big[ TV\big(\widehat{\mathcal{T}}(\cdot|z,a), \mathcal{T}(\cdot|z,a)\big) \Big] \tag{3}$$

This enables us to derive an upper bound on the regret that is independent of the reward function:

$$REGRET(\widehat{\pi}_R^*, \mathcal{M}^R) \leq \frac{4\gamma}{(1-\gamma)^2} \mathbb{E}_{z,a \sim d(\pi^{expl}, \widehat{\mathcal{M}})} \Big[ TV\big(\widehat{\mathcal{T}}(\cdot|z,a), \mathcal{T}(\cdot|z,a)\big) \Big] \text{ for all } R. \tag{4}$$

In the context of EX-BMDPs, the transition dynamics can be decomposed into two components: $\mathcal{T}(z'|z,a) = \mathcal{T}_+(s^{+'}|s^+, a)\mathcal{T}_-(s^{-'}|s^-)$, where $z = (s^+, s^-)$, $s^+$ represents the endogenous state, and $s^-$ represents the exogenous state. To apply EX-BMDP to visually distracted environments, we further assume that the reward depends solely on $s^+$, implying that decisions are made based only on endogenous state.

To model this separation, we construct the world model as two distinct components: the endogenous part $\widehat{\mathcal{M}}_+$ and the exogenous one $\widehat{\mathcal{M}}_-$, corresponding to the task-relevant and task-irrelevant processes, respectively. Their associated latent dynamics are denoted by $\widehat{\mathcal{T}}_+$ and $\widehat{\mathcal{T}}_-$. Utilizing this decomposition, we can derive a lower bound for the policy optimization target:

$$\begin{aligned} &\mathbb{E}_{z,a \sim d(\pi, \widehat{\mathcal{M}})} \Big[ TV\big(\widehat{\mathcal{T}}(\cdot|z,a), \mathcal{T}(\cdot|z,a)\big) \Big] \\ =&\mathbb{E}_{s^+, s^-, a \sim d(\pi, \widehat{\mathcal{M}})} \Big[ TV\big(\widehat{\mathcal{T}}_+(\cdot|s^+, a)\widehat{\mathcal{T}}_-(\cdot|s^-), \mathcal{T}_+(\cdot|s^+, a)\mathcal{T}_-(\cdot|s^-)\big) \Big] \\ \geq&\mathbb{E}_{s^+, a \sim d(\pi, \widehat{\mathcal{M}}_+)} \Big[ TV\big(\widehat{\mathcal{T}}_+(\cdot|s^+, a), \mathcal{T}_+(\cdot|s^+, a)\big) \Big] \end{aligned} \tag{5}$$

The strategy's optimization objective is a form of model uncertainty. Estimating model uncertainty for the URL serves as a reward for the exploration policy, which is crucial for learning. However, under the EX-BMDP assumption, directly maximizing the reward of the overall model error introduces significant bias. This is because the objective models not only the endogenous part but also the

exogenous part, while the strategy is solely based on $s^+$, leading to inefficient exploration. For the model optimization target, using triangle inequality, we have:

$$
\begin{aligned}
&\mathbb{E}_{z,a\sim d(\pi,\widehat{\mathcal{M}})}\Big[\mathrm{TV}\Big(\widehat{\mathcal{T}}(\cdot|z,a),\mathcal{T}(\cdot|z,a)\Big)\Big] \\
&\leq \mathbb{E}_{s^+,a\sim d(\pi,\widehat{\mathcal{M}}_+)}\Big[\mathrm{TV}\Big(\widehat{\mathcal{T}}_+(\cdot|s^+,a),\mathcal{T}_+(\cdot|s^+,a)\Big)\Big] \\
&\quad + \mathbb{E}_{s^-\sim d(\widehat{\mathcal{M}}_-)}\Big[\mathrm{TV}\Big(\widehat{\mathcal{T}}_-(\cdot|s^-),\mathcal{T}_-(\cdot|s^-)\Big)\Big]
\end{aligned}
\tag{6}
$$

With these bounds, we can reformulate the problem as a bi-level optimization problem:

**Problem 3.3.** *(Bi-level Optimization of World Model Error) Consider a reward-free EX-BMDP with latent dynamics functions $\widehat{\mathcal{T}}_+$, $\widehat{\mathcal{T}}_-$ and an exploration policy $\pi$, assuming the reward depends only on the endogenous state. Let $\mathcal{T} = \mathcal{T}_+ \times \mathcal{T}_-$ denote the true latent dynamics. We use a bi-level optimization: the inner optimization finds the world model minimizing the error, while the outer optimization maximizes the endogenous error with respect to the exploration policy.*

$$
\begin{aligned}
\text{Inner:}\quad & \min_{\widehat{\mathcal{T}}_+}\mathbb{E}_{s^+,a\sim d(\pi,\widehat{\mathcal{M}}_+)}\Big[\mathrm{TV}\Big(\widehat{\mathcal{T}}_+(\cdot|s^+,a),\mathcal{T}_+(\cdot|s^+,a)\Big)\Big] \\
& + \min_{\widehat{\mathcal{T}}_-}\mathbb{E}_{s^-\sim d(\widehat{\mathcal{M}}_-)}\Big[\mathrm{TV}\Big(\widehat{\mathcal{T}}_-(\cdot|s^-),\mathcal{T}_-(\cdot|s^-)\Big)\Big] \\
\text{Outer:}\quad & \max_{\pi}\mathbb{E}_{s^+,a\sim d(\pi,\widehat{\mathcal{M}}_+)}\Big[\mathrm{TV}\Big(\widehat{\mathcal{T}}_+(\cdot|s^+,a),\mathcal{T}_+(\cdot|s^+,a)\Big)\Big]
\end{aligned}
\tag{7}
$$

Problem 3.3 optimizes the world model error as a surrogate for Problem 3.1. It seeks a world model with low prediction error for both endogenous and exogenous components, guided by an exploration policy that maximizes endogenous error. This ensures the optimal policy generalizes well across varying exogenous distributions, regardless of the future reward function. Next, we introduce a practical framework to solve Problem 3.3.

# 4 Practical Implementation of SeeX

## 4.1 Control with Separation-assisted Latent Dynamics

**Separated World Model.** For high-dimensional inputs like images and videos, model-based frameworks encode past experiences into latent representations to predict future sequences [59, 65, 19]. Building on this, we propose a separated world model to disentangle task-relevant (endogenous) and task-irrelevant (exogenous) information. As shown in Figure 2, the endo-encoder $h_t^+ = E_{\theta+}(o_t)$ and exo-encoder $h_t^- = E_{\theta-}(o_t)$ extract these components. Two transition models handle dynamics: $q_{\theta+}(s_t^+|s_{t-1}^+,a_{t-1})$ for task-relevant states and $q_{\theta-}(s_t^-|s_{t-1}^-)$ for task-irrelevant ones. Additionally, inference models $p_{\theta+}(s_t^+|s_{t-1}^+,a_{t-1},h_t^+)$ and $p_{\theta-}(s_t^-|s_{t-1}^-,h_t^-)$ provide posterior estimates.

**Observation Model.** Some model-based approaches [20, 49] use an auxiliary reconstruction loss to refine the observation encoder, ensuring $z_t$ captures sufficient information from $o_t$ by maximizing the mutual information $\mathcal{I}(o_t;z_t)$. The IM algorithm [6] provides the Barber-Agakov lower bound: $\mathcal{I}(o_t;z_t) \geq \mathbb{E}_{p(o_t,z_t)}[\ln q_\theta(o_t|z_t)]+\mathcal{H}(p(o_t))$, where $\mathcal{H}(p(o_t))$ is the entropy of the observation distribution. In our separation setting, this bound becomes $\mathcal{I}(o_t;s_t^+,s_t^-) \geq \mathbb{E}_{p(o_t,s_t^+,s_t^-)}[\ln q_\theta(o_t|s_t^+,s_t^-)]$. Inspired by [63, 18, 24], We design an observation model $q_\theta(o_t|s_t^+,s_t^-)$, an endo-decoder $(\hat{o}_t^+,m_t^-) \sim D_{\theta+}(s_t^+)$, an exo-decoder $(\hat{o}_t^-,m_t^-) \sim D_{\theta-}(s_t^-)$ and a mask-mixing model $m_t = M_\theta(m_t^+,m_t^-)$. $\hat{o}_t^+$ and $\hat{o}_t^-$ are expected to reconstruct the task-relevant and task-irrelevant parts of the original observation, while $m_t^+$ and $m_t^-$ are corresponding masks. So $q_\theta(o_t|s_t^+,s_t^-)$ can be implemented as $\hat{o}_t = m_t \odot \hat{o}_t^+ + (1-m_t) \odot \hat{o}_t^-$. Considering insights from [18, 64] that task-relevant information occupies only a small portion of the observation, we design an exogenous reconstruction (Exo-Rec) model $\hat{o}_t^{\text{exo}} \sim q_\theta(o_t|s_t^-)$ to ensure that $s^-$ contains the vast majority of information.

**World Model Optimization.** Similar to a variational autoencoder (VAE) [30, 51], all model components are trained jointly by maximizing the evidence lower bound (ELBO). Our optimization

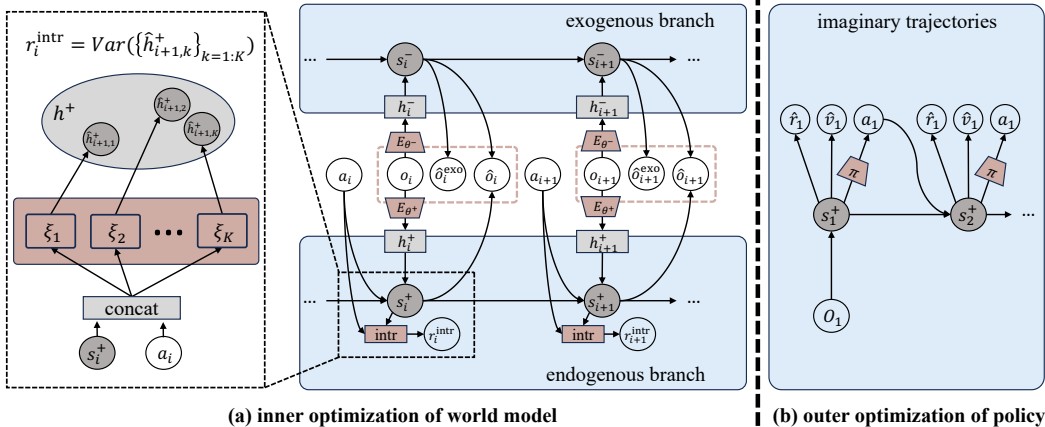

(a) inner optimization of world model     (b) outer optimization of policy

Figure 2: **(a)** Our separated world model comprises task-relevant and task-irrelevant branches, with ensemble predictive heads providing reward signals. **(b)** Imaginary trajectories within the endogenous branch enhance sample efficiency by reducing real-world interactions.

objective extends that of Dreamer [20]: $\mathcal{L} \doteq \mathbb{E}_p(\sum_t(\mathcal{L}_O^t + \mathcal{L}_R^t + \mathcal{L}_{KL}^t))$. The detailed formulations of each term are presented as follows, and the derivations are shown in Appendix B.1.

$$\mathcal{L}_O^t \doteq \ln q_\theta(o_t|s_t^+, s_t^-) + \textcolor{red}{\alpha} \ln q_\theta(o_t|s_t^-) \tag{8}$$

$$\mathcal{L}_R^t \doteq \ln q_\theta(r_t|s_t^+) \tag{9}$$

$$\mathcal{L}_{KL}^t \doteq \mathrm{KL}(p_{\theta+}(s_t^+|s_{t-1}^+, a_{t-1}, h_t^+)||q_{\theta+}(s_t^+|s_{t-1}^+, a_{t-1})) + \mathrm{KL}(p_{\theta-}(s_t^-|s_{t-1}^-, h_t^-)||q_{\theta-}(s_t^-|s_{t-1}^-, a_{t-1}))$$

**Policy Optimization.** To enhance sample efficiency, as shown in Figure 2-(b), we apply the policy optimization strategy of [20] to learn a parametric policy with imaginary trajectories. More specifically, we design an actor $\pi(a_t|s_t^+)$ and a reward model $q_\theta(r_t|s_t^+)$. The reward model fits the true reward function in the FT phase. The actor selects action based on the current endogenous state $s_t^+$ under the guidance of intrinsic reward in the PT phase or the reward model in the FT phase.

## 4.2 Intrinsic Reward: Compute Endogenous Uncertainty under Separation View

Crafting an effective intrinsic reward function is paramount during the pretraining (PT) phase. A well-designed intrinsic reward can steer the agent towards exploring states with the highest uncertainty, maximizing skill acquisition. A common practice [46, 47, 54] is to maximize the mutual information $\mathcal{I}(h_{1:T}^+;\xi|s_0^+, \pi_{\text{expl}})$, where $\xi$ represents the true unknown dynamics. This objective is motivated by the fact that mutual information quantifies how comprehensively the trajectory explores the environment.

**Proposition 4.1.** *(Single step's mutual information) If only we find the policy $\pi_{expl}$ that maximize every single step's mutual information $\sum_{t=0}^{T-1} \mathcal{I}(h_{t+1}^+;\xi|s_t^+, a_t)$, then the whole trajectories's mutual information $\mathcal{I}(h_{1:T}^+;\xi|s_0^+, \pi_{expl})$ will be also maximized. Proof in Appendix B.2.*

With Proposition 4.1, the optimization objective can be written as:

$$\pi_{\text{expl}}^* \doteq \arg\max_{\pi_{\text{expl}}} \sum_{t=0}^{T-1} \mathcal{I}(h_{t+1}^+;\xi|s_t^+, a_t) \tag{10}$$

$$= \arg\max_{\pi_{\text{expl}}} \sum_{t=0}^{T-1} \mathcal{H}(h_{t+1}^+|s_t^+, a_t) - \sum_{t=0}^{T-1} \mathcal{H}(h_{t+1}^+|\Xi = \xi, s_t^+, a_t) \tag{11}$$

To approximate the unknown $\xi$, we design a set of predictive heads $\{q_k(\hat{h}_{t+1,k}^+|\xi_k, s_t^+, a_t)\}_{k=1:K}$ which are implemented as conditional Gaussians $\mathcal{N}(\mu(\Xi = \xi_k, s_t^+, a_t), \sigma^2)$. Given fixed variance, the conditional entropy does not depend on state or action [54]. Suppose that $p(\Xi)$ is an uniform distribution on $\{\xi_k\}_{k=1:K}$, then we have $\mathcal{H}(h_{t+1}^+|s_t^+, a_t) = \frac{1}{K}\sum_{k=1}^{K} \mathcal{H}(h_{t+1}^+|\Xi = \xi_k, s_t^+, a_t)$. Since the

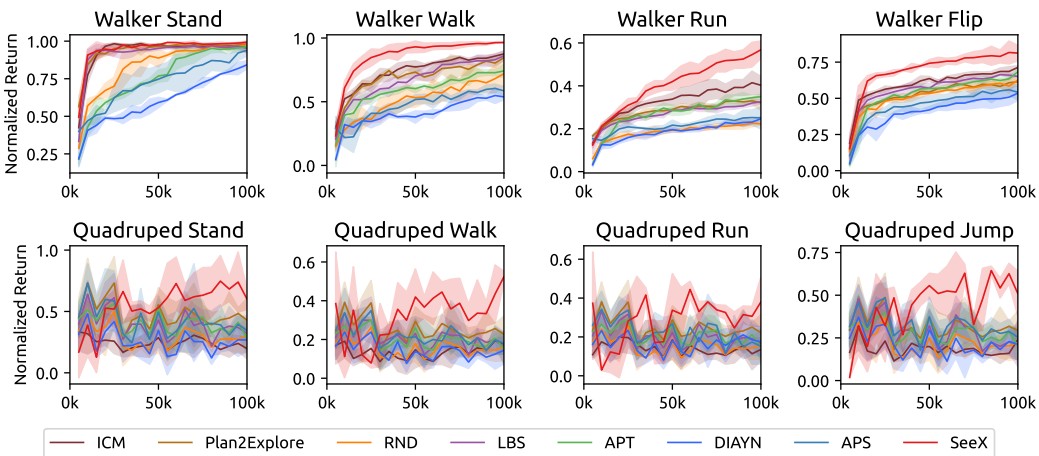

Figure 3: We show fine-tuning (FT) performance curves of SeeX and baselines across two domains and eight tasks. Pre-training (PT) used 2M frames, FT 100K. Normalized returns are benchmarked against the expert baseline [33], with mean (solid line) and variance (shaded area).

marginal entropy in Equation (11) lacks a closed-form expression amenable to optimization, we employ the empirical variance across ensemble means as a substitution. It is also an approximation to the world model TV error in Problem 3.3. We define the intrinsic reward as $r_t^{\text{intr}} = \text{Var}(\{\hat{h}_{t+1,k}^+\}_{k=1:K})$, and employ the exploratory policy $\pi_{\text{expl}}$ that maximizes $r_t^{\text{intr}}$ as an approximation of $\pi_{\text{expl}}^*$ in Equation (10).

## 5 Experiments

All experiments are conducted with at least three seeds and evaluated for 10 episodes. We conduct experiments to answer the following questions: (1) Can SeeX outperform other counterparts in URLB? (2) How do pre-training steps affect final performance? (3) Can SeeX give a reasonable model uncertainty estimation? (4) How do different components affect the model training? (5) Can the pre-trained world model and policy generalize to OOD distractors? (6) Can data augmentation handle visual distractors?

**Environments.** Following the URLB evaluation, we select three domains: *Walker*, *Quadruped*, and *Jaco Arm*, spanning twelve downstream tasks: Walker (stand, walk, run, flip), Quadruped (stand, walk, run, jump), and Jaco (reach-top-right, reach-top-left, reach-bottom-right, reach-bottom-left). These tasks vary in difficulty, providing a well-rounded performance assessment. The Jaco Arm domain is particularly challenging due to its multi-joint structure and sparse rewards, only granted upon catching the red ball. Visualizations of the environments are provided in Appendix A. Agents receive only visual inputs (64, 64, 3), with episodes lasting 1000 frames and an action repeat of $R = 2$. Pre-training utilizes up to 2M frames, and fine-tuning employs 100K frames, consistent with URLB-pixels.

**Datasets.** Leveraging the large-scale and high-quality Kinetics dataset [27], we construct two sub-datasets for our experiments: the **Driving-car** dataset, commonly used in previous works [18, 63], and the **Random-video** dataset, composed of videos randomly selected from other Kinetics classes. To evaluate generalization capabilities, we conducted experiments on the Random-video dataset Section 5.5 (*Note that we only use the Random-video here*). This zero-shot transfer assessment gauges the effectiveness of the policy on unseen random video scenarios.

**Baselines.** We choose seven different unsupervised methods as baselines, which can be categorized into three types: *Knowledge-based*: ICM [46], RND [10], LBS [43] and Plan2Explore [54] maximize prediction error to better understand the world; *Data-based*: APT [39] encourages exploration by maximizing entropy; *Competence-based*: DIAYN [16] and APS [38] maximize mutual information to achieve diverse discovery and generalization. The implementation detail and more concrete introduction of the above methods are shown in Appendix D.2.

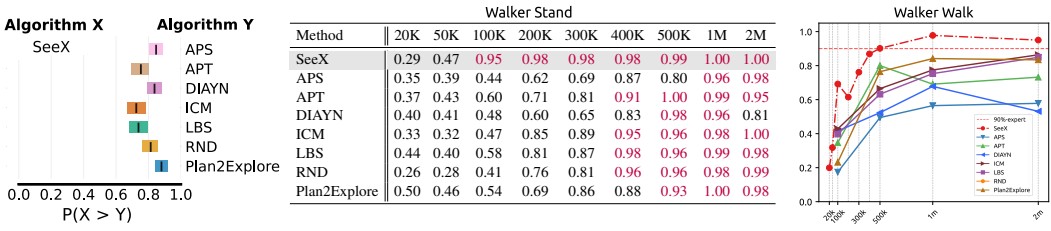

| Method | 20K | 50K | 100K | 200K | 300K | 400K | 500K | 1M | 2M |
|--------|-----|-----|------|------|------|------|------|-----|-----|
| SeeX | 0.29 | 0.47 | 0.95 | 0.98 | 0.98 | 0.98 | 0.99 | 1.00 | 1.00 |
| APS | 0.35 | 0.39 | 0.44 | 0.62 | 0.69 | 0.87 | 0.80 | 0.96 | 0.98 |
| APT | 0.37 | 0.43 | 0.60 | 0.71 | 0.81 | 0.91 | 1.00 | 0.99 | 0.95 |
| DIAYN | 0.40 | 0.41 | 0.48 | 0.60 | 0.65 | 0.83 | 0.98 | 0.96 | 0.81 |
| ICM | 0.33 | 0.32 | 0.47 | 0.85 | 0.89 | 0.95 | 0.96 | 0.98 | 1.00 |
| LBS | 0.44 | 0.40 | 0.58 | 0.81 | 0.87 | 0.98 | 0.96 | 0.99 | 0.98 |
| RND | 0.26 | 0.28 | 0.41 | 0.76 | 0.81 | 0.96 | 0.96 | 0.98 | 0.99 |
| Plan2Explore | 0.50 | 0.46 | 0.54 | 0.69 | 0.86 | 0.88 | 0.93 | 1.00 | 0.98 |

(a) Jaco Performance     (b) Walker-Stand/Walk performance vs. pretraining steps

Figure 4: **(a)** Average performance of four Jaco arm tasks. Each row shows comparative probabilities with 95% confidence intervals, indicating Algorithm X outperforms Algorithm Y [1]. Probabilities are based on 150 runs (50 per seed across 3 seeds) per task for robust evaluation. **(b)** Performance varies with pretraining steps. Given the small gap at 500k on Walker-Stand, a table highlights details, with red marking over 90% expert performance.

## 5.1    Evaluation on URLB with Distractors

The performance curves in Figure 3 show the normalized fine-tuning performance of SeeX and baseline methods pre-trained on 2 million frames. SeeX consistently outperforms or matches other methods across tasks, achieving expert-level performance in the walker-walk task. Notably, SeeX displays higher mean performance and greater variance, reflecting the positive correlation typical in reinforcement learning. In the quadruped domain, even SeeX's lowest performance exceeds the highest curves of other methods, a trend seen across quadruped tasks. Plan2Explore shows relatively weaker results, especially in the walker-stand and walker-walk tasks, likely due to relying solely on predictive heads based on the entire latent space. In contrast, our method demonstrates significant performance improvements. Other baselines like RND and APT perform reasonably well in simpler tasks but struggle with more complex tasks like walker-run and walker-flip. This indicates that using prediction errors, entropy maximization, or mutual information alone is inadequate. In contrast, our separated world model effectively extracts relevant information to enhance policy training.

## 5.2    Can SeeX Give a Reasonable Model Uncertainty Estimation?

As stated in Section 4.2 , model uncertainty is an approximation to the world model TV error in Problem 3.3. Thus we evaluate the accuracy of dynamics fitting for SeeX and Plan2Explore (both incorporating model uncertainty) with pre-trained models of 1M frames. To ensure a comprehensive assessment, we employ two policies: a random policy and SeeX's exploration policy (1M frames). As illustrated in Table 1, consistently across policies and domains, $U(s^+)$ values are lower than $U(z)$ values. This indicates that SeeX achieves a more accurate estimation of true dynamics compared to Plan2Explore, potentially contributing to its superior performance relative to

| Policy | Method | Walker | Quadruped | Jaco |
|--------|--------|--------|-----------|------|
| random | $U(s^+)$ | **0.04±0.00** | **0.17± 0.00** | **0.13±0.00** |
| | $U(s^-)$ | 0.64±0.03 | 33.0±31.0 | 0.48±0.03 |
| | $U(z)$ | 0.31±0.01 | 1.49±0.09 | 0.40±0.01 |
| SeeX | $U(s^+)$ | **0.05±0.00** | **0.18±0.00** | **0.14±0.00** |
| | $U(s^-)$ | 0.65±0.03 | 45.0±50.0 | 0.52±0.03 |
| | $U(z)$ | 0.35±0.01 | 1.78±0.08 | 0.44±0.02 |

Table 1: Uncertainty estimation of SeeX and Plan2Explore on the Driving-car dataset. Using two policies, we collected 1000 distinct states and reported the mean and standard deviation of uncertainty. $U(s^+)$, $U(s^-)$, and $U(z)$ denote the uncertainty for SeeX (endogenous and exogenous states) and Plan2Explore (latent belief state), respectively.

other baseline methods. Furthermore, a notable observation is that $U(s^-)$ values significantly exceed $U(s^+)$ values. This suggests that the total environmental uncertainty remains unchanged; rather, most uncertainty is concentrated in $q_{\theta^-}$ and $p_{\theta^-}$ due to SeeX's effective capture of task-irrelevant distractor transitions.

## 5.3    How do pre-training steps affect final performance?

Building upon the promising results from the previous section, where SeeX outperformed seven baseline methods and achieved expert-level performance from [33] (trained in a distractor-free environment, representing the upper bound for our setting) in some tasks, we delve into an intriguing

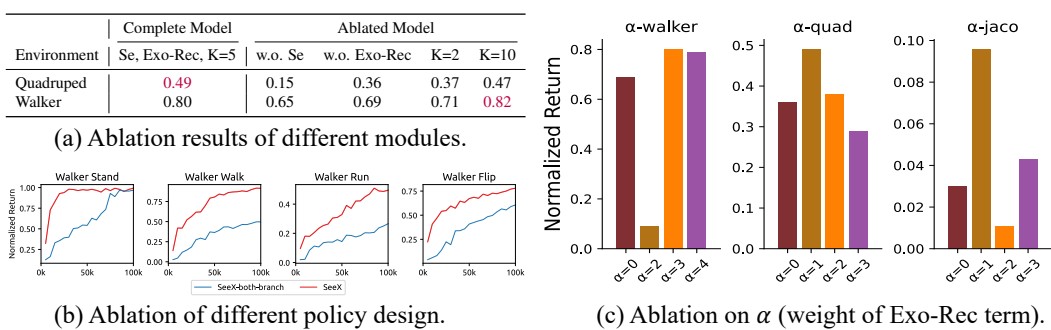

(a) Ablation results of different modules.

(b) Ablation of different policy design.

(c) Ablation on $\alpha$ (weight of Exo-Rec term).

Figure 5: Ablation results of SeeX with 500k fixed pretraining frames: **(a)** separation design, Exo-Rec term, and predictive head values; **(b)** different policy design: $\pi(s^+)$ (SeeX) and $\pi(s^+, s^-)$ (SeeX both-branch); **(c)** impact of different $\alpha$ (Exo-Rec weight) across three domains. Due to Jaco's task complexity, we ran 50 trials per seed and reported the top 30 mean returns.

question: how does the fine-tuning performance of SeeX vary with different pre-training step sizes? More specifically, can we achieve fine-tuned performance with as few pre-training steps as possible to achieve more than $90\%$ of expert performance? We conduct experiments on walker-stand and walker-walk. For all methods, we assessed the fine-tuning performance using pre-training frames of (100k, 500k, 1m, 2m). To evaluate SeeX on a more granular level, we add choices of (20k, 50k, 200k, 300k, 400k). As shown in Figure 4-(b), for the Walker-Walk task, SeeX reaches $90\%$ expert performance (red dotted line) within 500k frames. In contrast, no baseline achieves this level, even with 2M pre-training frames. For the simpler Walker-Stand task, a table offers detailed analysis. SeeX requires only 100k frames of random exploration to reach $90\%$ expert performance, while other baselines need significantly more pretraining.

## 5.4 How do different components affect the model training?

**Ablation of modules.** We first examine the impact of various modules in SeeX, including the separate world model, Exo-Rec design, and the number of predictive heads ($K$). As shown in Figure 5-(a), removing either the separate world model or Exo-Rec significantly reduces performance, underscoring the effectiveness of these designs. A small value of $K$ hinders accuracy, while performance improves with more heads, confirming our intuition. To balance estimation accuracy and computational cost, we choose $K = 5$.

**Ablation of policy.** In SeeX, we use only the endogenous state $s^+$ to predict actions. To validate this design, we compare two policies: $\pi(s^+)$ (SeeX) and $\pi(s^+, s^-)$ (SeeX-both-branch). As shown in Figure 5-(b), SeeX consistently outperforms SeeX-both-branch, particularly in the last three challenging tasks. This demonstrates that relying solely on endogenous information is beneficial in environments with moving distractors.

**Ablation of $\alpha$.** We conducted experiments to assess the impact of Exo-Rec's weight $\alpha$. As shown in Figure 5-(c), the optimal value of $\alpha$ varies across domains. For Jaco and quadruped tasks, a smaller $\alpha$ (e.g., 1) enhances performance, whereas walker tasks benefit from a larger weight (e.g., 2 or 3) to effectively extract exogenous information into $s^-$.

## 5.5 Can the Pre-trained World Model and Policy Generalize to OOD Distractors?

In this subsection, we evaluate the effectiveness of SeeX in addressing the "generalize to distractors from other distributions (OOD distractors)" challenge. To assess generalization ability, we compare SeeX with other baselines on four different walker tasks and report the average normalized return. We pre-trained and fine-tuned agents on the driving-car dataset and evaluated their performance on both driving-car and random-video datasets. The reported results represent the average normalized return across four walker tasks, with 50 runs conducted for each seed. As shown in Figure 6-(b), SeeX outperforms baselines on test distractor datasets and shows minimal performance drop under distribution shifts, highlighting its strong generalization.

| Method | Design | Stand | Walk | Run | Flip |
|---|---|---|---|---|---|
| SeeX | - | 0.99 | 0.91 | 0.49 | 0.78 |
| | fDA | **1.00** | **0.96** | **0.53** | **0.81** |
| | pDA | 0.97 | 0.89 | 0.45 | 0.71 |
| | pDA+fDA | 0.97 | 0.92 | 0.36 | 0.70 |
| Plan2Explore | - | 0.97 | 0.77 | 0.29 | 0.56 |
| | fDA | 0.98 | 0.81 | 0.33 | 0.61 |
| | pDA | 0.95 | 0.49 | 0.23 | 0.50 |
| | pDA+fDA | 0.95 | 0.56 | 0.24 | 0.53 |

| Method | Driving-car | Random-video | Drop($\downarrow$) |
|---|---|---|---|
| SeeX | **0.82** | **0.75$\pm$0.07** | **-0.09** |
| APS | 0.55 | 0.27$\pm$0.06 | -0.51 |
| APT | 0.64 | 0.55$\pm$0.07 | -0.14 |
| DIAYN | 0.63 | 0.34$\pm$0.08 | -0.46 |
| ICM | 0.69 | 0.56$\pm$0.07 | -0.19 |
| LBS | 0.70 | 0.61$\pm$0.08 | -0.13 |
| Plan2Explore | 0.69 | 0.58$\pm$0.06 | -0.16 |
| RND | 0.64 | 0.49$\pm$0.06 | -0.23 |

(a) Impact of data augmentation techniques.      (b) Generalization ability to OOD distractors.

Figure 6: **(a)** The impact of DA on performance in PT (pDA) and FT (fDA) stages. **(b)** Generalization ability to distractions from other distributions. The rightmost column indicates the percentage drop in performance caused by the distribution shift.

## 5.6 Can data augmentation handle visual distractors?

Numerous studies [3, 73, 70, 42] explore data augmentation (DA) in RL as an effective strategy for visual generalization. This subsection highlights the differences between DA and our approach. Our bi-level separation framework extracts task-relevant information, similar to how DA captures task-relevant representations. However, since few approaches use DA for exploration, we integrated it into Plan2Explore and SeeX to assess its impact on performance in the moving distractor setting. Specifically, we applied the classic random shift augmentation from [73] (4-pixel padding) four times per image. To evaluate DA's effectiveness, we tested it during both pretraining (pDA) and finetuning (fDA) on the walker task. The results in Figure 6-(a) reveal: (1) fDA improves performance by mitigating distractors. (2) SeeX's separation design outperforms fDA on tasks with moving distractors. (3) pDA reduces performance, likely by disrupting world model learning, further investigation is planned.

## 6 Related Work

**Model-based Control.** Learning a dynamics model of the environment is a promising way to tackle the problem of low sample efficiency, and has achieved impressive results in various tasks including continuous control as well as discrete control. To ensure the fairness of comparison, we adopted the official version of Plan2Explore and the implementation of URLB-pixels, which combines all baseline methods into a unified Dreamer-like framework. Plan2Explore makes use of the dreamerv2 framework and takes the uncertainty of prediction of the next latent state as the intrinsic reward. In our work, we focus on the scenarios with complex visual distractors and design an intuitive separated world model to capture the exogenous and endogenous states respectively, and learn policy with imaginary trajectories in latent space.

**Unsupervised RL.** In recent years, there are many works that tried to promote the unsupervised representation learning manner in various fields including computer vision (CV) [12, 23, 25] and natural language processing (NLP) [9, 48, 14]. Additionally, there have been numerous works applying RL algorithms to real-world applications, such as [37, 68, 44]. These works encourage the RL community to explore the more efficient way of learning [34, 32, 53, 58, 72, 11, 26], however, these works still need to optimize an extrinsic reward. Recently, there have been works to adopt a pure unsupervised manner (reward-free pre-training followed by reward-specific fine-tuning), which can be categoried into three kinds. (i) *Data-based*: maximal entropy RL has enabled agent for diverse exploration and data [39, 55, 71]; (ii) *Knowledge-based*: increase knowledge about the world with self-supervised prediction [46, 47]; (iii) *Competence-based*: methods based on mutual information [16, 22, 38, 57] shows capability for diverse discovery and generalization. URLB-pixels offers a unified model-based framework to implement some methods referred to above, we make use of this benchmark to serve as the baseline. A shared limitation of existing unsupervised exploration methods lies in their exclusive reliance on state information for exploration, lacking the ability to explicitly extract endogenous information. This inherent limitation renders these methods susceptible to performance degradation when confronted with visually rich environments containing complex distractors. Different from the above methods, our method designs an intuitive separated model to separate exogenous and endogenous information and only collect imaginary trajectories in

endogenous latent space to train policy. As a result, we largely improved performance compared to the other methods.

**Learning with Noisy Observations.** To address the issue of learning with noisy observations, diverse approaches have emerged, broadly categorized into four main strategies: (1) Use data augmentation methods to mitigate exogenous noises [72, 17, 74, 7]; (2) Learning task-relevant representations with bisimulation metrics [75, 40]; (3) Design auxiliary tasks to extract endogenous information [4, 5, 15, 31]; (4) Take actions or rewards as discriminating factors to separate exogenous and endogenous information [18, 64, 45, 41]. Our proposed method, SeeX, falls into the intersection of the last two categories, combining the benefits of auxiliary task design and discriminating factors to effectively handle noisy observations in complex real-world environments.

# 7 Conclusion

In this paper, we consider the problem of learning a policy in the visual unsupervised reinforcement learning (URL) setting with moving distractors. To tackle this intricate scenario, we introduce SeeX, a bi-level optimization framework that leverages a separated world model and task-relevant uncertainty maximization to mitigate the impact of distractors and enhance exploration efficiency. Furthermore, Our theoretical analysis formalizes URL with distractors: the policy maximizes task-relevant uncertainty to drive exploration, while the world model minimizes environmental uncertainty to reduce distractor influence. SeeX utilizes a separated world representation to disentangle exogenous and endogenous factors from the original observation domain. Policy training is then conducted exclusively on imaginary trajectories generated within the endogenous latent representation. Extensive experiments on the DMC-suite benchmark demonstrate that SeeX outperforms other baseline methods. Ablation studies provide insights into the contributions of individual components to our final performance.

**Limitations and Future Work.** Our work presents several areas for improvement: **(I)** Our work focuses on DMC tasks, leaving real-world applications like self-driving and navigation challenges for future exploration. **(II)** Distractors fall into four types based on their impact on rewards and actions: task-irrelevant + action-independent (our focus), task-relevant + action-independent, task-irrelevant + action-dependent, and task-relevant + action-dependent. The latter three will be addressed in future work. **(III)** Using only $s^+$ for policy learning is beneficial in our setting. However, for the task-relevant + action-independent case (e.g., multi-agent systems), we propose incorporating some $s^-$ into policy optimization as a potential solution.

## Acknowledgments and Disclosure of Funding

This work was partially supported by the National Science and Technology Major Project under Grant No. 2022ZD0114805, Collaborative Innovation Center of Novel Software Technology and Industrialization, NSFC (62376118, 62006112, 62250069, 61921006), and the Postgraduate Research & Practice Innovation Program of Jiangsu Province.

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

## A  Environment

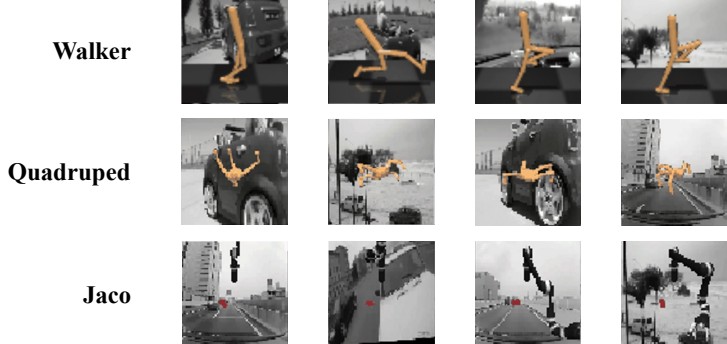

Figure 7:   We present visualization results for the three domains we employed, following the introduction of driving car distractors. Notably, we retain the floor for Walker while removing the floor for Quadruped. This is due to the fact that Quadruped's floor occupies the entire observation space, precluding the addition of moving distractions without floor removal.

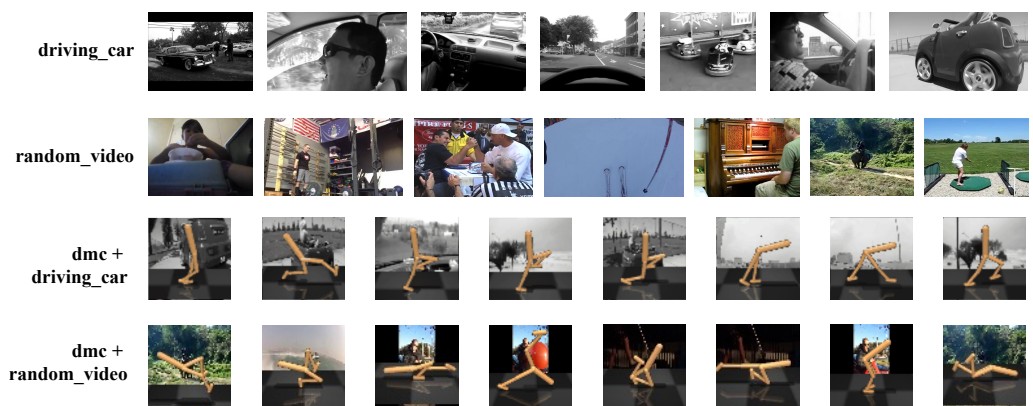

Figure 8:  To facilitate a clear understanding of the differences between the Driving-car and Random-video datasets, we present visualizations of both datasets.  Additionally, we showcase the DMC observations of various distractor types. A detailed description of the datasets employed can be found in Section 5.

To elucidate the direct impact of video distractors, we present illustrative visualizations in Figure 7. Various tasks present unique challenges.  For instance, the multi-jointed Jaco arm with sparse rewards (only awarded for catching the red ball) poses difficulties in accurately reconstructing its dynamics model, hindering the learning process. Additionally, the absence of a physical floor and the introduction of complex distractor videos in quadruped tasks make it challenging for the agent to determine its relative position without relying on the floor. Additionally, Figure 8 highlights the distinct characteristics of the Driving Car and Random Video datasets. Videos in Random-video feature RGB backgrounds that contrast sharply with the grayscale backgrounds in the Driving-car dataset.

## B  Proof

We base our approach on the following theoretical results from the paper.

## B.1 Optimization objective of dynamics model

We define an information bottleneck objective [61] to optimize the latent dynamics models same as many previous works [20, 21, 49, 63]. The objective can be written as:

$$\max \underbrace{\mathbb{I}(z_{1:T}; (o_{1:T}, r_{1:T})|a_{1:T})}_{(a)} - \beta \underbrace{\mathbb{I}(z_{1:T}, i_{1:T}|a_{1:T})}_{(b)} \tag{12}$$

Note that $\beta$ is a scalar and $i_t$ denotes indices of the dataset such that $p(o_t|i_t) = \delta(o_t - o'_t)$ [2]. We constrain the capacity of information contained in latent state $z_{1:T}$, while require them to predict observations and rewards as accurate as possible. We can lower bound term (a) by the non-negativity of the KL divergence:

$$\mathbb{I}(z_{1:T}; (o_{1:T}, r_{1:T})|a_{1:T}) = \mathbb{E}_{p(o_{1:T}, r_{1:T}, z_{1:T}, a_{1:T})}[\ln p(o_{1:T}, r_{1:T}|z_{1:T}, a_{1:T}) - \underbrace{\ln p(o_{1:T}, r_{1:T}|a_{1:T})}_{\text{model-irrelevant}}] \tag{13}$$

$$\overset{\pm}{=} \mathbb{E}_{p(o_{1:T}, r_{1:T}, z_{1:T}, a_{1:T})}[\ln p(o_{1:T}, r_{1:T}|z_{1:T}, a_{1:T})] \tag{14}$$

$$\geq \mathbb{E}_{p(o_{1:T}, r_{1:T}, z_{1:T}, a_{1:T})}[\ln p(o_{1:T}, r_{1:T}|z_{1:T}, a_{1:T})] \tag{15}$$

$$- \text{KL}\left(p(o_{1:T}, r_{1:T}|z_{1:T}, a_{1:T}) || \prod_{t=1}^{T} q(o_t|z_t)q(r_t|z_t)\right) \tag{16}$$

$$= \mathbb{E}_{q(z_{1:T}|o_{1:T}, a_{1:T})}\left[\sum_{t=1}^{T} \ln q(o_t|z_t) + \ln q(r_t|z_t)\right] \tag{17}$$

$$= \sum_{t=1}^{T}\left[\mathbb{E}_{q(s_t^+|o_{1:t}, a_{1:t-1})q(s_t^-|o_{1:t})} \ln q(o_t|s_t^+, s_t^-) + \ln q(r_t|s_t^+)\right] \tag{18}$$

The second term of Equation (13) can be dropped because the marginal data probability is irrelevant to the dynamics model $\widetilde{\mathcal{M}}$. Equation (18) is obtained by the assumption of EX-BMDP that the latent state $z_t$ can be decoupled into endogenous part $s_t^+$ and exogenous part $s_t^-$.

For term (b), we can directly make use of the derivation in SeMAIL [63] for its Equation (3), that

$$\mathbb{I}(z_{1:T}, i_{1:T}|a_{1:T}) \leq \sum_{t=1}^{T} \mathbb{E}_{q(s_{t-1}^+|o_{1:t-1}, a_{1:t-2})q(s_{t-1}^-|o_{1:t-1})} \tag{19}$$

$$\left(\text{KL}(p_{\theta^+}(s_t^+|s_{t-1}^+, a_{t-1}, s_t^+)||q_{\theta^+}(s_t^+|s_{t-1}^+, a_{t-1})) + \text{KL}(p_{\theta^-}(s_t^-|s_{t-1}^-, s_t^-)||q_{\theta^-}(s_t^-|s_{t-1}^-, a_{t-1}))\right) \tag{20}$$

## B.2 Proof of Proposition 4.1

**Proposition 4.1 Restated.** *(Single step's mutual information) If only we find the policy $\pi_{expl}$ that maximize every single step's mutual information $\sum_{t=0}^{T-1} \mathcal{I}(h_{t+1}^+; \xi|s_t^+, a_t)$, then the whole trajectories's mutual information $\mathcal{I}(h_{1:T}^+; \xi|s_0^+, \pi_{expl})$ will be also maximized.*

*Proof.*

$$\mathcal{I}(h_{1:T}^+; \xi|s_0^+, \pi_{expl}) = \sum_{t=1}^{T} \mathcal{I}(h_t^+; \xi|h_{t-1}^+, h_{t-2}^+, ..., h_1^+, s_0^+, \pi_{expl}) \tag{21}$$

$$= \sum_{t=1}^{T} \mathcal{I}(h_t^+; \xi|h_{t-1}^+, h_{t-2}^+, ..., h_2^+, s_1^+ \sim p_\theta(\cdot|s_0^+, \pi_{expl}(s_0^+), h_1^+), \pi_{expl}) \tag{22}$$

$$= \sum_{t=1}^{T} \mathcal{I}(h_t^+; \xi|s_{t-1}^+, \pi_{expl}) \tag{23}$$

$$= \sum_{t=0}^{T-1} \mathcal{I}(h_{t+1}^+; \xi|s_t^+, a_t) \tag{24}$$

Equation (21) is derived from the Chain Law of Mutual Information. Since we have already established the exploration policy and endogenous inference model $p_\theta$, we can sample the endogenous state $s^+$ for the next step. By iterating this process, we accumulate sufficient information to infer the endogenous state $s^+_{t-1}$, as described in Equation (23).

## C  Additional results

### C.1  Visualization of observational trajectories

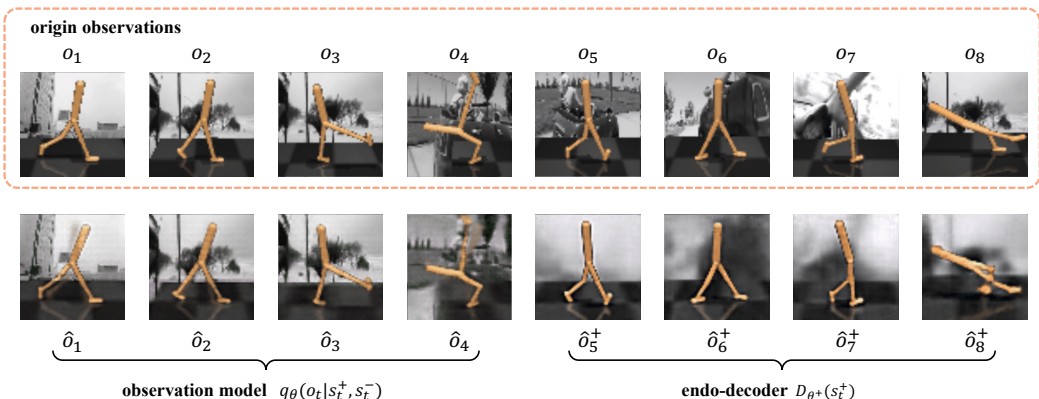

Figure 9:  Presentation of reconstruction results from the observation model and endo-decoder. The first row shows the original observations, while $\hat{o}_{1:4}$ represents the output of the observation model and $\hat{o}^+_{5:6}$ represents the output of the endo-decoder. Each corresponding index pair indicates a one-to-one relationship between the original observation and the reconstructed observation.

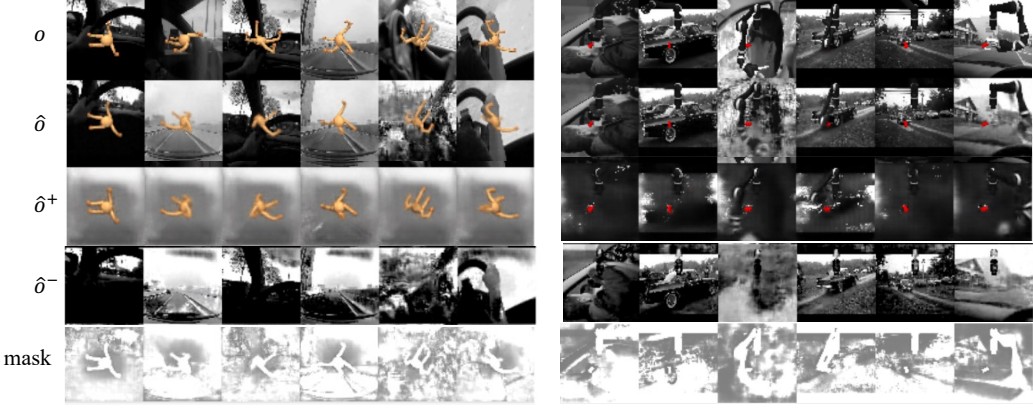

Figure 10:  We present the following: the original observation $o$, images reconstructed with both the endogenous and exogenous branches $\hat{o}$, images reconstructed using only the endogenous branch $\hat{o}^+$, images reconstructed using only the exogenous branch $\hat{o}^-$, and images with masking applied.

As illustrated in Figure 9, the observation model effectively reconstructs the original observations, including both background distractors and the agent. This indicates that $s^+_t$ and $s^-_t$ capture sufficient information from the observations. Furthermore, the endo-decoder outputs successfully reconstruct the endogenous information while eliminating exogenous distractors, demonstrating the ability of our separated model to distinguish task-relevant information from task-irrelevant information, enabling the agent to make decisions without being influenced by extraneous factors.

In Figure 9, we have shown the reconstruction trajectories of SeeX after fine-tuning in the Walker environment, but there was a lack of trajectory visualization of other environment during the exploration phase. Therefore, Figure 10 displays the reconstruction trajectories of SeeX during exploration in the Quadruped and Jaco environments. The trajectories are organized into five columns: from top to bottom, they represent the original observation $o$; the reconstruction $\hat{o}$ obtained from the joint use of $s^+$ and $s^-$ with the help of the mask $m$; the endogenous reconstruction $\hat{o}^+$ obtained using only $s^+$, representing the task-relevant part; the exogenous reconstruction $\hat{o}^-$ obtained using only $s^-$, representing the task-irrelevant moving distractors; and the mask $m$ used to reconstruct the entire image. By analyzing the reconstruction trajectories, we can draw the following conclusions:

- Although moving distractors were not completely removed, the reconstruction results show that $s^+$ contains most of the task-relevant information (including the agent's torso), significantly reducing the impact of background noise.

- During the exploration phase, we use intrinsic rewards to encourage the agent to explore a more diverse range of states, aiming to build as comprehensive a world model as possible. This foundation supports imagination during policy training in the fine-tuning phase. The reconstruction results indicate that the agent does not stick to fixed actions but attempts to explore diverse states, demonstrating the effectiveness of the intrinsic rewards used.

- The Jaco agent is a multi-joint robot with high flexibility, making it more complex than both the Walker and Quadruped. The reconstruction quality for Jaco is noticeably worse than for Quadruped, and the removal of moving distractors is less effective, which aligns with the experimental results in our paper. However, some interesting observations can be made: (1) The red ball, while not part of the agent's torso, is related to the reward function. It appears in $\hat{o}^+$ but not in $\hat{o}^-$, demonstrating SeeX's ability to automatically identify task-relevant information. (2) The base of Jaco, though fixed and not controlled by actions, is part of the agent's torso but has little relation to the reward. Therefore, SeeX classifies it as task-irrelevant, corresponding to the reconstruction image $\hat{o}^-$.

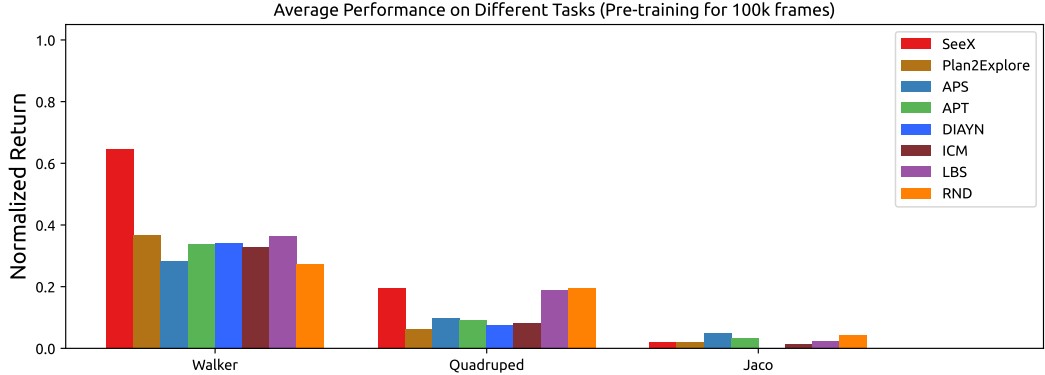

Figure 11: Fine-tuning performance comparison between SeeX and baselines, with 100k pre-training frames. Each bar represents the average normalized return across four walker tasks within a specific domain. We reported the mean of the top 30 highest returns.

## C.2 Complete pre-training performances

Owing to space constraints, the complete pre-training results are provided in the appendix. Figure 11, Figure 12, Figure 13, Figure 14 presents the fine-tuned performance of pre-trained models trained with varying pre-training frame durations.

# D Implementation Details

## D.1 Networks and Hyper-parameters

Building upon the recurrent state space model (RSSM) architecture [20, 54], we retain the forward dynamics, posterior encoder, and policy components from the implementation in [54]. The hyper-

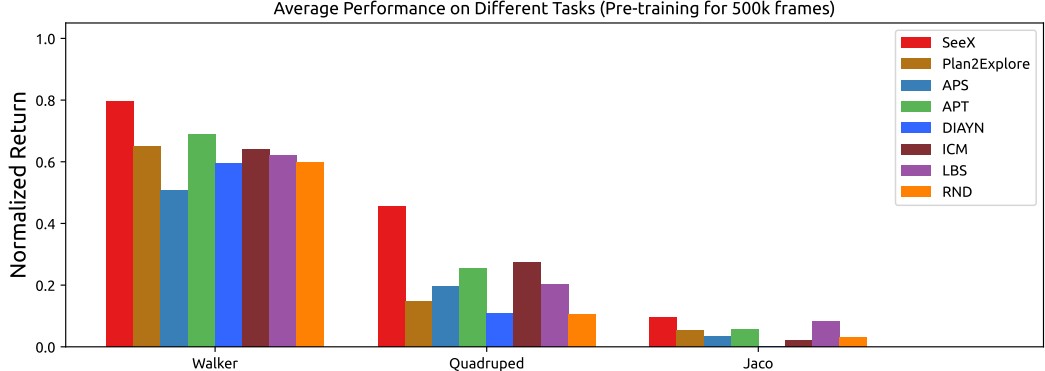

Figure 12: Fine-tuning performance comparison between SeeX and baselines, with 500k pre-training frames. Each bar represents the average normalized return across four walker tasks within a specific domain. We reported the mean of the top 30 highest returns.

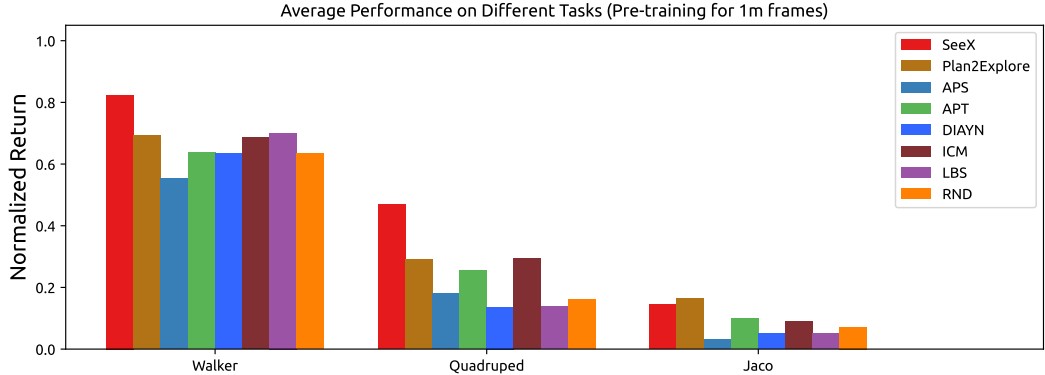

Figure 13: Fine-tuning performance comparison between SeeX and baselines, with 1m pre-training frames. Each bar represents the average normalized return across four walker tasks within a specific domain. We reported the mean of the top 30 highest returns.

parameters for SeeX are detailed in Table 2. We introduce a masked image encoder, a novel image decoder for reconstructing the original observation, and an additional RSSM to model the transition of exogenous states.

Elaborating further, the exo-encoder $E_{\theta-}$ mirrors the structure of the endo-encoder $E_{\theta+}$ (i.e., the image encoder in [54]). However, we introduce modifications to the conventional decoder to construct our novel endo-decoder $D_{\theta+}$ and exo-decoder $D_{\theta-}$, as expressed in the following:

- 1 fully connected layer with 1536 hidden dimensions.
- Reshape tensor into shape of $(-1, 1536, 1, 1)$.
- 4 transposed convolution layers with 6 output channels, stride 2, and ELU activation with $\alpha = 1.0$.
- Decompose 6-channel input into two 3-channel structures. The first represents the reconstruction output, while the remaining three channels encode the corresponding mask.
- 1 output transformation layer of Gaussian distribution.

To effectively reconstruct the observed state, we introduce a novel network architecture for the observation model $q_\theta(o_t|s_t^+, s_t^-)$. This network integrates the outputs of the two decoders described earlier to generate the final reconstruction. The detailed network architecture is as follows:

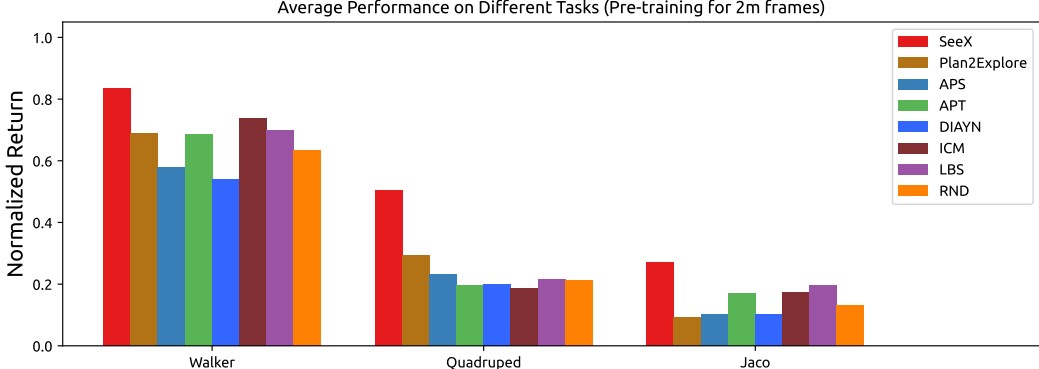

Figure 14: Fine-tuning performance comparison between SeeX and baselines, with 2m pre-training frames. Each bar represents the average normalized return across four walker tasks within a specific domain. We reported the mean of the top 30 highest returns.

- Get two images and two masks from $D_{\theta+}$ and $D_{\theta-}$ as input.
- 1 2D convolution layer with 3 output channels and kernel size 1. Input: two masks, Output: a final mask.
- Weight two images with the final mask as weights.
- Output the final image.

| Hyperparameter | Value |
|---|---|
| $\alpha$ of Exo-Rec | 3 for Walker, 1 for Quadruped and Jaco |
| KL weight | 1 for $s^+$ and $s^-$ |
| Deterministic size | 32 |
| Stochastic size | 32 |
| Discrete | 32 |
| Embedding size | 200 |
| Sequence length $T$ | 50 |
| Batch size | 50 |
| Imagine horizon $H$ | 15 |
| Precision | 16 |
| Model optimizer | Adam: {lr: 3e-4, eps: 1e-5, clip: 100, wd: 1e-6} |
| Actor optimizer | Adam: {lr: 8e-5, eps: 1e-5, clip: 100, wd: 1e-6} |
| Critic optimizer | Same as actor's |
| Action repeat | 2 |

Table 2: Hyperparameters of SeeX for all experiments.

## D.2 Detailed Description about Baselines

**ICM.** Curiosity is quantified as the discrepancy between the predicted consequences of the agent's actions and the actual outcomes, as represented in a learned visual feature space. This formulation enables efficient exploration, outperforming baseline methods in VizDoom and Super Mario Bros. Notably, ICM [46] exhibits superior exploration capabilities even in the absence of explicit external rewards. The study underscores the significance of evaluating generalization to new scenarios and offers valuable insights into the effectiveness of reinforcement learning algorithms in adapting to novel environments.

**Plan2Explore.** During exploration, Plan2Explore [54] effectively formulates plans to seek out novel experiences, enabling it to rapidly adapt to downstream tasks in a zero-shot or few-shot setting.

Surpassing existing self-supervised exploration techniques, Plan2Explore achieves competitive zero-shot task performance and eventually outperforms a supervised agent in few-shot scenarios. By leveraging unsupervised exploration to learn a world model, Plan2Explore demonstrates remarkable scalability and data efficiency, paving the way for significant advancements in building real-world reinforcement learning systems.

**RND.** By leveraging a neural network to predict features of observations extracted from a randomly initialized network, RND [10] effectively enhances exploration in Atari games characterized by sparse rewards. Notably, the method surpasses state-of-the-art performance on challenging games like Montezuma's Revenge, achieving remarkable results without relying on human demonstrations or access to the game's underlying state. The paper underscores the flexibility of combining intrinsic and extrinsic rewards, demonstrating that RND facilitates directed exploration at a local level but faces challenges in global exploration over extended time horizons. Overall, the RND method represents significant progress in tackling hard exploration tasks, highlighting the potential of simple yet effective techniques at scale.

**LBS.** Latent Bayesian Surprise (LBS) [43] leverages Bayesian surprise in a latent space to generate intrinsic rewards for exploration in Reinforcement Learning (RL). LBS surpasses existing methods on a diverse range of tasks, demonstrating efficient exploration in both continuous-control and discrete-action environments. It exhibits remarkable resilience to stochasticity in the environment dynamics, facilitating in-depth exploration and achieving superior performance in high-dimensional settings such as video games. Notably, LBS significantly reduces computational costs while enhancing exploration capabilities, making it a promising method for enhancing RL algorithms.

**APT.** APT [39] actively acquires behaviors and representations by exploring novel states in reward-free environments. This approach maximizes non-parametric entropy in an abstract representation space, circumventing the need for complex density modeling and enabling effective scaling to high-dimensional observation environments. Leveraging intrinsic motivation and particle-based entropy maximization, APT achieves human-level performance on Atari games and surpasses benchmark results on DMControl tasks. Future research directions include reducing sample complexity through model-based RL integration and investigating methods to mitigate catastrophic forgetting during fine-tuning. Overall, the paper showcases APT's efficacy in enhancing performance on challenging RL tasks while requiring substantially fewer samples compared to fully supervised RL algorithms.

**DIAYN.** By maximizing an information-theoretic objective using a maximum entropy policy, DIAYN [16] enables the unsupervised emergence of diverse skills, such as walking and jumping, in simulated robotic tasks. The method demonstrates remarkable effectiveness in exploring complex environments, often solving benchmark tasks without receiving explicit task rewards. Additionally, DIAYN offers strategies for rapid task adaptation, hierarchical reinforcement learning, and imitation learning. By maximizing the mutual information between states and skills, DIAYN enhances the empowerment of a hierarchical agent and ensures skill diversity through maximum entropy policies. Overall, the paper's contributions include proposing a novel method for unsupervised skill learning, showcasing the emergence of diverse skills in various tasks, demonstrating adaptability to new tasks, and providing a foundation for hierarchical reinforcement learning and imitation tasks. DIAYN presents a promising approach for enhancing exploration and data efficiency in reinforcement learning settings.

**APS.** By seamlessly integrating variational successor features with nonparametric entropy maximization, APS [38] effectively optimizes the mutual information between tasks and policy-induced states. This method outperforms existing benchmarks on the Atari 100k data-efficiency benchmark by employing entropy maximization to navigate the environment and leveraging data for behavioral learning. To stabilize training and improve convergence, the method incorporates averaging over k nearest neighbors. APS addresses the shortcomings of previous methods by maximizing the entropy of policy-induced states in a lower-dimensional abstract representation space. Empirical results on the Atari benchmark demonstrate state-of-the-art performance, showcasing substantial progress over earlier work. The paper highlights the benefits of utilizing state entropy maximization data for task-conditioned skill discovery and outlines future research directions for further optimization and the integration of diverse approaches.

## D.3 Experiments Compute Resources

We implement SeeX with Pytorch and run all the experiments on NVIDIA RTX 3090 for about 7500 GPU hours. The pre-training phase takes 10G of memory and 80 GPU hours for 2M frames, while the fine-tuning phase takes 10G of memory and 6 GPU hours for 100k frames. The codes of SeeX can be found in the supplementary materials.

