# OpenReview forum: "Leveraging Separated World Model for Exploration in Visually Distracted Environments"
_NeurIPS.cc/2024/Conference — NeurIPS 2024 poster_

### Official Review · Reviewer_NjAE · 2024-06-25

**Soundness:** 2
**Presentation:** 1
**Contribution:** 2
**Rating:** 4
**Confidence:** 4

**Summary:**

To address the challenge of visual distractions in unsupervised reinforcement learning, this paper proposes a bi-level optimization framework called SeeX, which utilizes a separate world model to mitigate the disturbance caused by visual distractions. The authors evaluate the proposed method in multiple tasks.

**Strengths:**

- While unsupervised RL relies on intrinsic reward to promote exploration, the existence of visual distractions might contaminate the evaluation of intrinsic reward. SeeX proposes to use a separate world model to mitigate such disturbance.

**Weaknesses:**

- The separation model assumes decoupled endogenous and extraneous transition, which could be a strong assumption. For example, in robot manipulation, obstacles might be irrelevant to reward signals but still affect the movement of the robot.
- The separation model uses a mask model to combine endogenous and extraneous reconstruction. However, this might not work well when the distractions and embodiments are inseparable, e.g. Color Change or Camera Jittering [1].
- The proposed EX-BMDP seems to be similar to Task-Informed MDP in [2], while the proposed intrinsic reward is similar to the latent disagreement in [3], I have some reservations about the novelty of combining these two methods as they are generally orthogonal.

[1] Denoised MDPs: Learning World Models Better Than the World Itself

[2] Learning Task Informed Abstractions

[3] Planning to Explore via Self-Supervised World Models

Typos:
- Line 51we use the task-irrelevant (-> task-relevant) information to predict the reward and the action.

**Questions:**

- Line 193: "It is also an approximation to the world model TV error in Problem 3.3". Can the author provide more explanation on why Eq (11) is consistent with Eq (7)?
- The authors wish to have a separate world model. However, how do you ensure that $s^-$ does not contain information about the reward? Since you have $\hat o_t^{exo}$?
- In the experiment, what RL algorithm are the baselines implemented with?
- In Eq (6), the REGRET is bounded by the estimation error of $\hat{\mathcal{T}}^+$ and $\hat{\mathcal{T}}^-$. However, if $s^-$ is irrelevant to reward signals and endogenous state $s^+$, and $\pi$ only depends on $s^+$, why would the estimation error of $\hat{\mathcal{T}}^-$ affect the REGRET?

---

> ### Author Rebuttal · Authors · 2024-08-07
>
> Thank you for your review and feedback. The term "task-irrelevant" on L51 should be "task-relevant". Here are our responses to your questions:
> - **Strong assumption**
>   - Firstly, many Visual RL works use separation assumptions, such as Denoised MDP[3] and TIA[2], which have natural assumptions and practical significance. SeeX is inspired by this line of research. We believe no single assumption is universally applicable (e.g., CNNs for images and RNNs for text). So whether an assumption is "strong" is subjective and context-dependent.
>   - Secondly, we focus on URL problems with task-irrelevant and action-independent noise and we model URL as minmax problems through theoretical analysis. A **practical scenario** fitting our assumption is a book-finding robot in a library. The robot must differentiate between task-relevant items (e.g., books) and task-irrelevant items (e.g., posters or people) to find the specific book. Thus, our work is grounded in both theory and practical application.
>   - Finally, the robot manipulation you mentioned is discussed in L331-334.
>
> - **Inseparable noises**
>   - We emphasized in the paper (caption of Figure 1 and the conclusion) that we only focus on handling moving distractors. Moving distractors and inseparable noises represent **distinct research directions**. According to the No Free Lunch theorem, no single method can be effective for all tasks.
>
> - **Discussion on whether EX-BMDP and latent disagreement are orthogonal**
>   - **Separated world model**
>     - Firstly, we **adopted**(L78-79) rather than **proposed** EX-BMDP[1]. The similarity you see between EX-BMDP and TIA is due to their shared use of the separation concept, **but separation is not a specific method but rather a general idea**[1,2,3,4].
>     - Additionally, EX-BMDP differs significantly from TiMDP[2]. TiMDP applies action to the exogenous branch, while EX-BMDP does not. Did the reviewer notice this? The agent can only influence the endogenous branch under our assumption, which is why EX-BMDP is suitable.
>   - **Intrinsic reward**
>     - As we all know, during pretraining phase of unsupervised RL, task-specific rewards are typically avoided in favor of intrinsic rewards, which come in various forms[5,6,7,8]. **Any intrinsic reward can support SeeX's exploration**, and we chose the classic disagreement[5].
>   - Our method formalizes URL as a minmax problem, where the outer layer minimizes REGRET’s upper bound using a separation assumption (Eq 2), leading us to adopt EX-BMDP. Inner layer maximize state uncertainty, so we approximate Eq 7 with Eq 11, making disagreement a reasonable choice. Reviewer considers the methods to be orthogonal, but we believe that the underlying ideas abstracted (e.g., separation and intrinsic reward) can inspire cross-field applications. Our step-by-step design and component choices are based on necessity, and our results validate the framework's effectiveness.
>
> - **L193, the relationship between Eq 7 and Eq 11**
>   - In Eq 7, the goal of $\pi_{expl}$ is to maximize the total variation (TV) distance between the learned model and the true latent dynamics in the task-relevant part. Practically, we estimate $D_{TV}$ using the disagreement among an ensemble of neural networks, inspired by [9].
>
> - **How to ensure $s^-$ does not contain information about the reward?**
>   - Firstly, our Exo-Rec design follows [2,3], which states that task-relevant information is a small part of the observation. The Exo-Rec term aims to encode as much information from $o$ into $s^-$ (L164-166), not to remove reward-related information from $s^-$ as you mentioned.
>   - Secondly, We assume that only the task-relevant part of $o$ is affected by the action. Based on EX-BMDP, we design a separated world model where the action limits the encoding of task-relevant information into $s^+$.
>   - In summary, **we believe there is a misunderstanding of our method and the Exo-Rec design**. In visual settings, reward-relevant and reward-irrelevant information are inherently mixed and cannot be perfectly separated. Are you asking us to ensure a 100% separation between the two? This is not feasible. Our results(Figure 3, Table 1) show that our method attempts to separate $s^-$ and $s^+$ as much as possible, benefiting the task of handling moving distractors.
>
> - **The implementation of baseline methods**
>   - We have mentioned in the paper (L47-49, L73-77, L201) that our method builds on URLB[10] and URLB-pixel[11], using the same implementation for the baseline. The specific implementation of the baseline can be found in the URLB-pixel GitHub repository.
>
> - **The REGRET bound in Eq (6)**
>   - According to your statement, if we extract **precise** task-irrelevant information ($\hat{T}^-$) from the observation, it wouldn’t help in reducing REGRET? This seems unreasonable because reward-relevant and reward-irrelevant information are complementary and together constitute the observation. Knowing one allows us to infer the other.
>   - Although $s^-$ is unrelated to REGRET, we use losses of $s^-$ and $s^+$ to bound REGRET (Eq 6). If you believe that $s^-$ is insignificant, you might try deriving a tighter REGRET bound based solely on $s^+$.
>
> We hope this clarifies our work. Please feel free to ask any further questions.
>
> [1] Provably Filtering Exogenous Distractors using Multistep Inverse Dynamics
>
> [2] Learning Task Informed Abstractions
>
> [3] Denoised MDPs: Learning World Models Better Than the World Itself
>
> [4] SeMAIL: Eliminating Distractors in Visual Imitation via Separated Models
>
> [5] Planning to Explore via Self-Supervised World Models
>
> [6] Curiosity-driven exploration by self-supervised prediction
>
> [7] Exploration by random network distillation
>
> [8] Curiosity-driven exploration via latent bayesian surprise
>
> [9] Reward-free curricula for training robust world models
>
> [10] URLB: Unsupervised Reinforcement Learning Benchmark
>
> [11] Mastering the unsupervised reinforcement learning benchmark from pixels

---

> ### Author Response · Authors · 2024-08-11
>
> Dear reviewer NjAE,
>
> Since the discussion phase deadline is approaching, we would like to send a friendly reminder.
>
> We greatly appreciate your time and dedication to providing us with your valuable feedback. We hope we have addressed the concerns, but if there is anything else that needs clarification or further discussion, please do not hesitate to let us know.
>
> Thanks, Authors

---

> > ### Comment · Reviewer_NjAE · 2024-08-12
> >
> > Thank the authors for the explanation, I have read the rebuttal and decided to keep my score.

---

> ### Author Response · Authors · 2024-08-12
>
> Thank you for reviewing our rebuttal. We are glad to hear that you do not have any new questions. If you have any further concerns or need clarification, please feel free to reach out at any time.

---

### Official Review · Reviewer_HYxU · 2024-07-10

**Soundness:** 2
**Presentation:** 2
**Contribution:** 2
**Rating:** 4
**Confidence:** 3

**Summary:**

This paper studies the problem of intrinsic-driving exploration in visually distracted environments, in the context of unsupervised reinforcement learning (URL). To address the issue that intrinsic rewards might be biased by distractors, the authors propose a method (called SeeX) that separates exogenous and endogenous information by training separate world models. In this way, the pre-trained world will be able to focus mainly on the task-relevant part of the observation, and the intrinsic reward based on dynamic uncertainty will be more accurate. The proposed method is evaluated on 3 image-based continuous control domains with video distractors. The results show that SeeX outperforms other URL methods on some environments.

**Strengths:**

* Separating exogenous and endogenous information to produce a better intrinsic reward is an interesting idea.
* The writing is mostly clear. The diagrams look nice.

**Weaknesses:**

(Not all points are weaknesses. Comments/questions/weaknesses are all gathered in this section for convenience.)

* Line 157: what is BA?
* Line 164: should the second $=$ be $+$?
* Figure 4b (walker stand): this figure is a bit misleading. It seems to suggest that SeeX is much better than other methods for pretrain budgets between 100k ~ 500k. However, this is not necessarily true, as there are no data points at 200k/300k/400k for baselines. The claim that other methods require **at least** 500k frames to achieve the same level of performance is ungrounded.
* Apart from Figure 9, could the authors provide more visualizations for the decoded observations? For example, $\hat{o}_t^-$, $m_t$, and trajectories with different embodiments.
* Have the authors conducted ablation experiments (K/Se/Exo-Rec) on other environments than Quadruped? Considering the quadruped results are quite noisy (Figure 3), it would be better to also include ablation results on walker envs.
* Ablation: how does SeeX perform when $\alpha=0$?

Other issues:
* The legend of Figure 5 is a bit confusing. It took me some time to understand that the bottom legend is for the bottom-right figure. Please consider rearranging its position.

**Questions:**

Please see weaknesses above.

**Limitations:**

The limitations have been briefly discussed in Section 7.

---

> ### Author Rebuttal · Authors · 2024-08-06
>
> Thank you for your detailed review. We have noted the typographical and formatting issues mentioned by the reviewer. BA on Line 157 refers to the Barber-Agakov bound[1], named using the authors' initials, which is a commonly used bound for mutual information. The second "=" on Line 164 should indeed be a "+". We believe these adjustments do not affect the main content and conclusions of the paper. Below are our responses to the issues you raised.
> - **Regarding conclusions of Figure 4**
>   - Our original intention was to show SeeX's performance stability between 100k and 500k steps; we sampled and tested the performance of other methods between 100k~500k, but none exceeded 90% expert performance, so we did not report them in the paper. We show the performance of all steps for the other methods in **Pdf-Table-B** and our refined conclusion is that SeeX achieves 90% expert performance with fewer pre-training steps than other methods, demonstrating the efficiency of our bi-level framework and the separated world model during exploration.
> - **Other trajectory plots except for Figure 9**
>   - In Figure 9, we have shown the reconstruction trajectories of SeeX after fine-tuning in the Walker environment, but there was a lack of trajectory visualization of other environment during the exploration phase. Therefore, **Pdf-Figure-C** displays the reconstruction trajectories of SeeX during exploration in the Quadruped and Jaco environments. The trajectories are organized into five columns: from top to bottom, they represent the original observation $o$; the reconstruction $\hat{o}$ obtained from the joint use of $s^+$ and $s^-$ with the help of the mask $m$; the endogenous reconstruction $\hat{o}^+$ obtained using only $s^+$, representing the task-relevant part; the exogenous reconstruction $\hat{o}^-$ obtained using only $s^-$, representing the task-irrelevant moving distractors; and the mask $m$ used to reconstruct the entire image. By analyzing the reconstruction trajectories, we can draw the following conclusions:
>     - Although moving distractors were not completely removed, the reconstruction results show that $s^+$ contains most of the task-relevant information (including the agent's torso), significantly reducing the impact of background noise.
>     - During the exploration phase, we use intrinsic rewards to encourage the agent to explore a more diverse range of states, aiming to build as comprehensive a world model as possible. This foundation supports imagination during policy training in the fine-tuning phase. The reconstruction results indicate that the agent does not stick to fixed actions but attempts to explore diverse states, demonstrating the effectiveness of the intrinsic rewards used.
>     - The Jaco agent is a multi-joint robot with high flexibility, making it more complex than both the Walker and Quadruped. The reconstruction quality for Jaco is noticeably worse than for Quadruped, and the removal of moving distractors is less effective, which aligns with the experimental results in our paper. However, some interesting observations can be made: (1) The red ball, while not part of the agent's torso, is related to the reward function. It appears in $\hat{o}^+$ but not in $\hat{o}^-$, demonstrating SeeX's ability to automatically identify task-relevant information. (2) The base of Jaco, though fixed and not controlled by actions, is part of the agent's torso but has little relation to the reward. Therefore, SeeX classifies it as task-irrelevant, corresponding to the reconstruction image $\hat{o}^-$.
> - **Regarding the ablation studies in Figure 5**
>   - When formatting Figure 5, we only considered aesthetic factors and overlooked the potential for misleading the reviewers. Therefore, we have re-drawn the conclusions of the ablation study and improved the readability of the charts, and added the ablation experiments on the walker (**Pdf-Table-C**) as well as the performance with $\alpha=0$ (**Pdf-Figure-B**). By analyzing the charts, we can draw the following conclusions:
>     - The choice of weight for the Exo-Rec term significantly impacts SeeX's performance. For Walker, selecting $\alpha=3$ is appropriate, while for Quadruped and Jaco, $\alpha=1$ is more suitable. Removing the Exo-Rec term (equivalent to setting $\alpha=0$) leads to varying degrees of performance degradation across different environments, indicating that the Exo-Rec design contributes positively to performance.
>     - Observing **Pdf-Table-C**, SeeX's sensitivity to different hyperparameters shows similar conclusions for Walker and Quadruped: (1) Removing the separated world model significantly impacts performance; (2) Removing the Exo-Rec term aligns with the conclusions from **Pdf-Figure-B**; (3) SeeX's performance is generally positively correlated with the number of predictive heads $K$, but considering the computational cost associated with increasing $K$, $K=5$ is a suitable choice.
>
> Thank you again for your support and constructive feedback on our work. We believe these improvements will further enhance the quality and impact of the paper.
>
> [1] The IM algorithm: a variational approach to information maximization

---

> ### Author Response · Authors · 2024-08-11
>
> Dear reviewer HYxU,
>
> Since the discussion phase deadline is approaching, we would like to send a friendly reminder.
>
> We greatly appreciate your time and dedication to providing us with your valuable feedback. We hope we have addressed the concerns, but if there is anything else that needs clarification or further discussion, please do not hesitate to let us know.
>
> Thanks, Authors

---

> > ### Comment · Reviewer_HYxU · 2024-08-12
> >
> > I apologize for my late reply. Thank you to the authors for addressing my questions and providing additional results. I have updated my score to 4, but I still have a few questions:
> >
> > * Regarding the additional ablation results on Walker, why was a table used instead of a line plot, as in the paper? Additionally, for the ablations related to $\alpha$, why not use a line plot similar to Figure 5 (top-left)? I also didn't understand why different plots were used for the ablation results on $\alpha$ in Jaco and Walker.
> > * I share similar reservations about the novelty as Reviewer NjAE. To me, the main takeaway from this paper is that using only task-related information to measure disagreement is a better approach. Describing this as a completely new framework is unconvincing. Furthermore, making statements like "Any intrinsic reward can support SeeX's exploration" without experimental results to support it can be misleading and overly generalizing.

---

> ### Author Response · Authors · 2024-08-12
>
> Thank you for your prompt and thoughtful response to our rebuttal. We appreciate the time and effort you’ve taken to review our rebuttal and provide additional feedback. Below are our responses to the new questions and concerns you raised:
> - **Additional ablation results on Walker**
>   - As you mentioned in your review, “there are no data points at 200k/300k/400k for baselines,” we understand your concern. To address this, we have included the results in a table, providing more specific quantitative values to better illustrate the performance comparison of different methods at various time steps. Considering your preference for the same type of qualitative line charts as before, we will update the new data points in the line charts of the later formal version.
> - **Ablation of $\alpha$​**
>   - Firstly, thank you for pointing out in your review that Figure 5 was somewhat confusing. In Figure 5, the top-left, top-right, and bottom-left plots show ablations with different values of $\alpha$ in various environments; since their content is consistent, we present them as bar charts in the PDF for a clearer comparison of their normalized returns. The bottom-right plot depicts ablation experiments of different modules, and we provide a more detailed analysis of these experiments in the table, which shows the specific numerical improvements.
>   - Secondly, to address your and future readers' confusion, in the later formal version, we will replace the previous inconsistent Figure 5 with a unified and clearer set of figures. Additionally, we will include the original line plots in the appendix for reference, so readers can still gain a qualitative understanding of the performance of the ablation experiments.
> - **Reservations about the novelty**
>   - First, thank you for considering that our approach of using only task-related information to measure disagreement is an improvement. Below, I will provide a more detailed explanation of our novelty and contributions.
>   - In real-world scenarios, there are many sources of action-independent and task-irrelevant noise. For example, when a drone is patrolling, surrounding buildings, or distant vehicles do not directly affect the drone's flight path or the completion of its patrol task. Many current works overlook this issue, and we are **the first** to address such noise in the URL setting. Additionally, we propose a bi-level optimization framework centered around task-related information and design our methods based on this theoretical foundation. Finally, our experimental results demonstrate the effectiveness of our approach, and ablation studies confirm the utility of different components.
> - **Statements without experimental results**
>   - Considering the URL problem setting, our model-based framework requires intrinsic reward to assist in early exploration. Intrinsic reward is a modular and can be implemented in various ways. For example, the disagreement measure $Var(\hat{h}^+_{t+1,i})$ used in our work and the variance of the log of predicted probabilities $Var(\log \hat{T}_i(z_t))$ used in [1], both of the intrinsic rewards can be used within our framework.
>   - Our main contribution is a theoretically-based framework for solving the URL problem with visual disturbances. In principle, intrinsic reward can be implemented based on the ideas from different methods in URLB[2], as well as the two specific approaches mentioned earlier, meaning our framework is **intrinsic-reward-agnostic**. As you mentioned, the experimental performance of these intrinsic rewards has indeed not been tested. We will consider conducting more extensive and comprehensive comparisons in future work.
>
> Thank you for your continued effort in providing multiple rounds of feedback to improve our work. Your questions have been very valuable. Please feel free to reach out with any further questions or concerns you may have in the future.
>
> [1] Offline Reinforcement Learning from Images with Latent Space Models
>
> [2] URLB: Unsupervised Reinforcement Learning Benchmark

---

> > ### Author Response · Authors · 2024-08-13
> >
> > We hope our responses address your concerns. If you have any further questions or need clarification, please do not hesitate to reach out to us.

---

> > > ### Comment · Area_Chair_ttVR · 2024-08-13
> > > **Response**
> > >
> > > Does the author response address your concerns? Please acknowledge and send any follow up questions.

---

> > > > ### Comment · Reviewer_HYxU · 2024-08-14
> > > >
> > > > Thank you for the reminder. I don't have further questions that require the authors to clarify.

---

> > > > > ### Author Response · Authors · 2024-08-14
> > > > >
> > > > > Thank you for your valuable feedback and for the effort you've put into helping us improve our work. We greatly appreciate the insightful discussions we've had with you throughout this process. Rest assured, we will incorporate these discussions and your suggestions into the later formal version of the paper. Your contributions have been instrumental in refining our work, and we are truly grateful for your time and consideration.

---

### Official Review · Reviewer_wcfo · 2024-07-13

**Soundness:** 3
**Presentation:** 3
**Contribution:** 2
**Rating:** 6
**Confidence:** 4

**Summary:**

This paper considers the problem of unsupervised reinforcement learning (task-agnostic pretraining) from image observations in environments with visual distractors. The key technical contribution of this paper is a practical algorithm, SeeX, based on the world model framework common in MBRL literature. SeeX explicitly separates endogenous (agent has control over) and exogenous (agent has no control over) forward dynamics in the learned world model, such that downstream policy optimization can occur only with the endogenous branch of the world model. The intuition behind the approach is that, assuming that the agent has no control over visual distractors (i.e., actions do not affect the distractors), we can expect such information to be encoded only in the exogenous branch of the model, thus not affecting policy optimization. The proposed method is validated on URLB, a common benchmark for unsupervised RL from images, and compared against prior work in this area.

**Post-rebuttal comment:** I believe that the authors have addressed my concerns and I am revising my score from 5 -> 6 and recommend acceptance.

**Strengths:**

- The problem is interesting and relevant to the NeurIPS community, as well as RL researchers and practitioners more broadly. I believe that the approach is fairly straightforward yet original.
- Paper is generally well written and easy to follow. There is sufficient background for an unfamiliar reader to appreciate the technical contributions.
- Theoretical analysis helps build an intuition for the proposed method. Experiments are well thought out and the proposed method is compared to a variety of strong baselines from the URLB. Results seem to indicate that the proposed method indeed benefits from its separate branches.

**Weaknesses:**

I do not have any major concerns with the paper in its current form. However, there's a couple of things that I had expected to see in the paper:
- A very common and effective strategy for visual generalization in RL has been use of data augmentation. There are numerous papers on the topic, many of which leverage simple and highly general augmentations such as random shift/crop, color jitter, mixup/overlay. Based on the strong similarity in problem setting, I would expect data augmentation to work very well on the considered benchmark. Intriguingly, data augmentation seems to have the same limitations as the proposed method, in the sense that they cannot capture distractors that the agent can interact with (i.e., has some control over). I would like the authors to comment on the similarity between the two research directions, and potentially include some discussion on this in the paper. I would strongly suggest including 1-2 data augmentation baselines for the camera-ready version of the paper as well; I fully acknowledge that this may not be a reasonable ask for the rebuttal itself.
- I believe that there is another limitation of the proposed method which I didn't see any mention of in the paper. As far as I understand, moving objects that are task-relevant but external to the agent may not be captured by the endogenous branch. For example, a multi-agent setting in which another agent may move independently, or perhaps an environment in which a task-relevant object is moving due to e.g. physics without the agent actively interacting with the object. Is my understanding correct? This seems somewhat distinct from the example given in L333.
- I understand why the authors would choose to only optimize the policy using the endogenous branch. However, I believe it would be quite valuable to include an ablation in which the policy is optimized with access to both branches; it is not obvious to me whether this would have any notable impact on downstream task performance.

**Questions:**

I would like the authors to address my comments in the "weaknesses" section above. While I do have concerns regarding baselines, I believe that strong verbal arguments and/or changes to the writing is sufficient to address my concerns during the rebuttal. However, I will expect the authors to follow through with any promises of additional experimental results or changes to the paper for a future camera-ready version.

**Limitations:**

The authors discuss limitations in L331-334. However, I would like the authors to elaborate on the limitations of their approach.

---

> ### Author Rebuttal · Authors · 2024-08-06
>
> Thank you for the detailed review and valuable suggestions regarding our paper. We are pleased to hear that you find our work to have clear technical contributions and to perform well in experiments. In response to the questions and suggestions you have raised, we provide the following answers:
> - **Comparison between our research direction and data augmentation (DA)**
>   - Our designed bi-level separation framework extracts task-relevant information, while the DA method extracts task-relevant representations through data augmentation. As you mentioned, there are indeed similarities between the two approaches. However, since there are few methods that directly use DA for exploration, we incorporated the DA method into both Plan2Explore and SeeX to investigate whether it enhances performance. This is to verify whether the DA method provides benefits in the moving distractor setting.
>   - We used the classic data augmentation technique of random shift (pad by 4 pixels) proposed in DrQ[1], applying it 4 times to the same image. To thoroughly evaluate the effectiveness of DA, we tested its impact during both the pretraining (p-DA) and finetuning (f-DA) stages on the walker task. The performance results are shown in **Pdf-Table-A**.
>   - By analyzing the data in the table, we can draw the following conclusions:
>     - Applying data augmentation (f-DA) on specific tasks is effective, as it can mitigate the impact of distractors and improve performance, which supports the reviewer's insight.
>     - However, for tasks with moving distractors, the separation design in SeeX performs better than data augmentation.
>     - Using data augmentation during the pretraining stage leads to a performance drop, possibly because it interferes with the learning of the world model. The specific reasons will be explored in future work.
> - **Further clarification on the content from L331-334**
>   - Based on whether they affect the agent's reward function and action, distractors can be categorized into four types: task-irrelevant + action-independent (the setting addressed in our work); task-relevant + action-independent (which I understand to be the multi-agent setting you mentioned); task-irrelevant + action-dependent; and task-relevant + action-dependent. The latter two are the settings I referred to on L331-334, where distractors can interact with agents and even influence their reward function.
>   - We believe the reviewer's understanding is correct and we appreciate you pointing this out. We will include the settings you mentioned in the camera-ready version.
> - **Ablation in which the policy is optimized with access to both branches**
>   - Firstly, by formalizing the URL problem as a minmax problem, we further implement it as a separated world model consisting of two branches: endogenous and exogenous. The exogenous branch extracts task-irrelevant information, which is unrelated to the task. Therefore, it is reasonable to use only the endogenous branch to optimize the policy.
>   - Secondly, I understand that you are referring to the ablation experiments for using both branches as inputs to the policy. To address your concerns comprehensively, we consider the following two scenarios: (1) using both exogenous and endogenous states simultaneously for policy optimization, and (2) merging the endogenous and exogenous branches into one, using a single encoder to extract the state $s$ for policy optimization.
>     - (1) We added the following comparative experiments: both $s^+$ and $s^-$ are simultaneously input to both the actor $\pi(a|s^+,s^-)$ and the critic $v(s^+,s^-)$, with the action still applied to the endogenous branch. We compared the performance of this setup (SeeX-both-branch) with SeeX in the walker environment, and the specific performance curves are shown in **Pdf-Figure-A**. In all tasks, SeeX outperformed SeeX-both-branch, especially in the last three challenging tasks. This indicates that using only endogenous information is advantageous for tasks with moving distractors.
>     - (2) Using a single encoder to encode observations for training the policy $\pi(a|z)$, which is the approach of Plan2Explore[2]. The superiority of SeeX over Plan2Explore is demonstrated in Figures 3 and 4 of our paper, showcasing the effectiveness of the separated world model.
>   - In summary, we believe that using only the endogenous branch for policy optimization is reasonable, and our experiments and the content in the paper have demonstrated this. Although using information from two branches under our assumption doesn't offer advantages, as noted by the reviewer, there are multi-agent settings where noise, despite action independence, can affect the agents' reward functions. In such cases, incorporating part of $s^-$ into policy optimization might be effective. This is an interesting research direction that we are glad to explore in future work.
>
> Thank you again for your support and constructive feedback on our work. We believe these improvements will further enhance the quality and impact of the paper. We will ensure that the additional experimental results and any necessary changes are included in the camera-ready version of the paper.
>
> [1] Image Augmentation Is All You Need: Regularizing Deep Reinforcement Learning from Pixels
>
> [2] Planning to Explore via Self-Supervised World Models

---

> > ### Comment · Reviewer_wcfo · 2024-08-08
> >
> > Thank you for the detailed response to my questions. I believe that the new experimental results and clarification addresses my main concerns, especially the ablation using both branches is helpful in motivating the approach. I have increased my score from 5 -> 6 with the expectation that the authors incorporate all reviewer feedback into a camera-ready version.

---

> > > ### Author Response · Authors · 2024-08-09
> > >
> > > Thank you very much for your thoughtful feedback and for taking the time to review our paper. We sincerely appreciate your constructive comments and are pleased to hear that the new experimental results and clarifications address your main concerns.
> > >
> > > We are especially grateful for your recognition of the ablation study involving both branches, and we will certainly incorporate all the feedback provided into the camera-ready version of the paper. Your input has been invaluable in helping us refine our work and ensure its quality.
> > >
> > > Thank you once again for your support and for increasing the score. We are committed to making the necessary revisions to meet the expectations and enhance the clarity and impact of our paper.

---

### Official Review · Reviewer_SZqk · 2024-07-31

**Soundness:** 4
**Presentation:** 4
**Contribution:** 3
**Rating:** 8
**Confidence:** 3

**Summary:**

The authors propose a method for separating endogenous and exogenous latent states for unsupervised exploration under visual distractors. Motivated by a theoretical bound minimizing the regret under a latent world model, the algorithm learns both an endogenous and exogenous world model, as well as an exploratory policy trained to maximize an intrinsic reward.

They evaluate on continuous control tasks with various distractors, showing their method outperforms other baselines. They perform additional experiments on how pretraining affects downstream finetuning RL performance and how their choice of intrinsic reward is a good approximation for the total variation found in the theoretical bound. They also provide analysis on parameter choices for their algorithm.

**Strengths:**

The algorithm is extremely well-motivated from theory and does a good job arguing that the assumptions made in theory match reality. The paper is clear and rich with useful insights and the experimental results are strong.

**Weaknesses:**

Since the method performed so well in these settings, it would have been nice to see how it performed on a task in the real world under realistic endo/exo genous conditions

**Questions:**

Can you see this method helping in something like a self driving or navigation task in the real world?

**Limitations:**

Yes, they discuss the important limitation that the endogenous and exogenous state are always fully separate and don't interact with one another.

---

> ### Author Rebuttal · Authors · 2024-08-06
>
> Thank you for your detailed review and valuable feedback on our work. We are very pleased that you found our algorithm theoretically motivated and experimentally successful. In response to your questions and suggestions, we have made the following points:
> - **Exploration of Real-World Applications**
>   - Our work focuses on handling moving distractors, where we need to separate task-relevant states from task-irrelevant states. In this setup, there are many common downstream tasks. For example, consider a book-finding robot in a library that needs to scan the shelves to find a specific book. The robot will encounter task-relevant objects (such as books) and task-irrelevant objects (such as posters on the wall or passing people), so the robot needs to distinguish which information is beneficial for the task.
>   - We agree with your view that real-world applications are crucial for validating the practicality of the algorithm. However, the self-driving and navigation challenges you mentioned are difficult to analyze and computationally expensive in practice. We would be willing to address these in future work. And this will involve designing experiments closer to real-world conditions to verify the algorithm's effectiveness in handling complex real-world scenarios.
> - **Discussion on Method Limitations**
>   - The assumption of completely separating endogenous and exogenous states that you mentioned is indeed a limitation of our method. We have discussed this in the paper, and our work has preliminarily validated the effectiveness of the bi-level optimization framework and the separated world model under the setting where endogenous and exogenous information is fully separable. In future work, we will explore how to introduce interactions between states to better understand and improve the applicability of the algorithm.
>
> Once again, thank you for your support and constructive feedback on our work. We believe these improvements will further enhance the quality and impact of the paper.

---

> ### Author Response · Authors · 2024-08-12
>
> Dear reviewer SZqk,
>
> Since the discussion phase deadline is approaching, we would like to send a friendly reminder.
>
> We greatly appreciate your time and dedication to providing us with your valuable feedback. We hope we have addressed the concerns, but if there is anything else that needs clarification or further discussion, please do not hesitate to let us know.
>
> Thanks, Authors

---

> > ### Comment · Reviewer_SZqk · 2024-08-12
> > **Rebuttal Response**
> >
> > I thank the authors for their response and discussion! I have no further questions at this time

---

> > > ### Author Response · Authors · 2024-08-13
> > >
> > > Thank you for your kind words and for taking the time to review our responses and discussion. I appreciate your positive feedback and will carefully consider any potential improvements to further enhance the quality of our paper. If you have any additional comments or suggestions in the future, please do not hesitate to reach out.

---

### Author Rebuttal · Authors · 2024-08-06

We thank the reviewers for taking the time to give useful comments for our paper. We are glad that the Reviewers appreciate the correct motivation (Reviewer SZqk), original approach (Reviewer wcfo) and strong experimental results (Reviewer SZqk, wcfo). Reviewers pointed out the concerns and points for improvement, and we made responses for each of them. We presented more experimental results and detailed responses to questions about our assumption and approach. \
We kindly request the reviewers to inform us if there is anything else we can clarify. Thank you for taking the time to review our work.

---

### Decision · Program_Chairs · 2024-09-25

**Decision:**

Accept (poster)

**Comment:**

This paper proposes a new unsupervised RL algorithm for environments with visual distractors. The key idea is to learn a separated world model to distinguish between task relevant information and task irrelevant distractors. There is agreement that the approach is sound, motivations are clear, has a good theoretical/experimental grounding.

The key strengths that were highlighted were: the approach is novel and well motivated, theoretical foundation and analysis is strong, the benchmark tasks are impressive, and the presentation is clear. Although there were some weaknesses reported as well: justification and validation for why certain assumptions were made, relationship to prior work and scope of evaluation outside of standard benchmarks, to more real world applications.

Reviewers answered detailed responses to questions raised during the review process, including ablation studies and clarifications. I recommend accepting this paper for a poster presentation.